# Diaphanous-related formin mDia2 regulates beta2 integrins to control hematopoietic stem and progenitor cell engraftment

Yang Mei [1,2✉], Xu Han[1,2], Yijie Liu[1,2], Jing Yang[1,2], Ronen Sumagin[1,2] & Peng Ji [1,2✉]

Bone marrow engraftment of the hematopoietic stem and progenitor cells (HSPCs) involves homing to the vasculatures and lodgment to their niches. How HSPCs transmigrate from the vasculature to the niches is unclear. Here, we show that loss of diaphanous-related formin mDia2 leads to impaired engraftment of long-term hematopoietic stem cells and loss of competitive HSPC repopulation. These defects are likely due to the compromised trans-endothelial migration of HSPCs since their homing to the bone marrow vasculatures remained intact. Mechanistically, loss of mDia2 disrupts HSPC polarization and induced cytoplasmic accumulation of MAL, which deregulates the activity of serum response factor (SRF). We further reveal that beta2 integrins are transcriptional targets of SRF. Knockout of beta2 integrins in HSPCs phenocopies mDia2 deficient mice. Overexpression of SRF or beta2 integrins rescues HSPC engraftment defects associated with mDia2 deficiency. Our findings show that mDia2-SRF-beta2 integrin signaling is critical for HSPC lodgment to the niches.

[1] Department of Pathology, Feinberg School of Medicine, Northwestern University, Chicago, IL 60611, USA. [2] The Robert H. Lurie Comprehensive Cancer Center, Northwestern University, Chicago, IL 60611, USA. ✉email: yang.mei@northwestern.edu; peng-ji@fsm.northwestern.edu

Hematopoietic stem and progenitor cells (HSPCs) reside in bone marrow in adults where stromal cells provide a favorable microenvironment for their proliferation and differentiation[1,2]. Under certain circumstances, cytokine-stimulated HSPCs migrate through the bone marrow–blood barrier and enter the peripheral blood. The ability of HSPCs to migrate and engraft allow bone marrow transplantation (BMT) to be used as a routine clinical strategy to treat patients with various hematologic diseases. After transplantation, the donor HSPCs rapidly localize to the central marrow region, where they redistribute to produce preferential seeding of long-term hematopoietic stem cells (LT-HSCs) that are close to the endosteal region[3,4]. After this homing process, long-term engraftment of HSPCs eventually leads to repopulation of various hematopoietic lineages. Although previous studies revealed critical factors involved in HSPC homing and engraftment[5], how these cells transmigrate from the vasculatures to the niches is unclear.

Regulation of the membrane cytoskeleton network by actin polymerization has been demonstrated to be critical for HSPC homing, migration, and engraftment[6,7]. In mammals, there are two major actin-nucleating protein families: the Wiskott–Aldrich syndrome protein (WASP)-Arp2/3 protein complexes and diaphanous-related formin proteins that nucleate branched and unbranched F-actin filaments, respectively[8]. The diaphanous-related formins, including mDia1, 2, and 3, are characterized by a unique and highly conserved actin polymerization formin homology 2 (FH2) domain that is preceded by a proline-rich FH1 domain. These two enzymatic domains are flanked by the N-terminal Rho GTPase-binding domain (GBD) and the C-terminal diaphanous auto-regulatory domain (DAD). The FH1 domain, in a complex with profilin, recruits G-actin subunits to the FH2 domain for polymerization of linear actin filaments. The diaphanous-inhibitory-domain (DID) and the dimerization domain (DD) are required for subcellular localization of mDia fomins[9]. Biochemical studies showed that the DID and DAD domains interact to form an auto-inhibitory structure[8,10]. This auto-inhibition of mDia formins is further supported by findings showing that mDia formin variants with mutations in the DAD or N-terminal regions are constitutively active[11–14]. When activated Rho GTPase binds to the GBD domain, the auto-inhibitory loop is relieved and mDia formins are activated[8]. mDia formins can also be activated through post-translational modifications[15].

In many cell types, mDia formins are involved in diverse processes including cytokinesis, cell polarity, endocytosis, filopodium formation, adhesion, and migration[8,16]. The dynamics of cell adhesion and motility are tightly regulated by actin cytoskeleton rearrangements. This regulation is particularly important for the activity of various hematopoietic cells, such as immune cell migration in response to antigens and HSPC engraftment. Although WASP is known to be required for HSPC migration and engraftment[17], whether mDia formins are involved in these processes is unclear. Several studies have shown that mDia formins play important functions in hematopoietic cells including T cells[18,19], neutrophils[20–22], macrophages[23], and erythroid cells[24,25]. Many of these studies used knockout mouse models to reveal the functions of mDia in vivo. mDia1-deficient mice are viable, but develop symptoms that mimic age-associated human myelodysplastic syndromes[26,27]. Mice with global knockout of mDia2 are embryonically lethal with defects in fetal erythropoiesis[28]. We recently generated a mDia2 conditional knockout mouse model. Mice with hematopoietic-specific knockout of mDia2 exhibit anemia with many bi-nucleated late-stage erythroblasts and significant defects in terminal erythropoiesis in the bone marrow[25]. In this study, we use these mouse models to reveal that mDia2 plays a critical role in HSPC engraftment. In particular, we find that loss of mDia2 does not compromise

HSPC homing to the bone marrow vasculatures. Instead, mDia2 is essential for transendothelial migration from the vasculature and subsequent lodging in bone marrow niches. We further reveal the presence of a mDia2–MAL–SRF–beta2 integrin signaling axis that mediates mDia2 function in HSPC bone marrow lodging.

## Results

**Loss of mDia2 does not affect steady-state HSPC compositions.** The mDia formin proteins in mice include mDia1, mDia2, and mDia3 that are encoded by *Diap1, 3*, and *2*, respectively. Prior studies demonstrated that mDia2 is enriched in the hematopoietic cells[24,29]. Using a quantitative real-time PCR, we confirmed that mDia2 is highly expressed in HSPCs (Supplementary Fig. 1a). By analysis of the Gene Expression Commons database, we also found that mDia2 is enriched in HSPC populations (Supplementary Fig. 1b).

To determine the role of mDia2 in HSPC function, we first analyzed the content of HSPCs in a mDia2$^{fl/fl}$Vav-Cre mouse model in which mDia2 deletion specifically in hematopoietic tissue initiated on embryonic day 12[30]. The HSPCs we analyzed include LSKs (lineage negative, Sca-1+, c-Kit+), LKs (lineage negative, c-Kit+, Sca1−), multipotent progenitors (MPPs, LSK, CD34+, CD135+), long-term HSCs (LT-HSCs LSK, CD34−, CD135−), short-term HSCs (ST-HSCs, LSK, CD34+, CD135−), and SLAM (CD150+, CD48−, LSK) HSCs as we performed previously[26,27]. These mice showed anemia as reported[25]. However, there were no significant changes on HSPC contents in the bone marrow (Supplementary Fig. 1c, d). In addition, cell cycle analysis showed minimal changes in HSPC quiescence in mDia2-deficient mice (Supplementary Fig. 1e, f). Together, these results indicate that mDia2 may not be required to maintain the steady state of HSPCs, which prompted us to focus on the role of mDia2 in HSPC homing and engraftment during BMT.

**mDia2 is required for HSPC engraftment in transplantation.** BMT represents a stress condition for HSPCs. We first performed a transplantation assay in which CD45.2+ bone marrow mononuclear cells (BMMC) from control or mDia2$^{fl/fl}$ Vav-Cre mice were transplanted non-competitively into lethally irradiated CD45.1+ recipient mice. Compared to mice transplanted with wild-type (WT) control bone marrow cells, mice transplanted with mDia2 null cells were anemic, which is consistent with our previous report[25]. When we analyzed the HSPCs 4 months after transplantation, LT-HSC and ST-HSC populations were proportionally increased (Fig. 1a, b) and a trend toward increases in the absolute cell number of the HSPCs in the bone marrow was observed (Fig. 1c).

Analysis of the cell cycle status of the LSK population in recipient mice transplanted with BMMCs from mDia2$^{fl/fl}$ Vav-Cre mice showed cells increased in G1 but decreased in G0 phases (Fig. 1d, e). This result suggests a loss of quiescence in mDia2-deficient LSK cells after BMT, which could lead to the initial expansion but later exhaustion of HSPCs. Indeed, when we analyzed these mice 10 months after transplantation, we observed significantly reduced LSK and LS cells (Supplementary Fig. 2a) and a continued loss of quiescence (Supplementary Fig. 2b). These phenotypes were also observed in the pIpC-treated mDia2$^{fl/fl}$Mx-Cre mouse model, an inducible system that enables hematopoietic-specific mDia2 knockout in adult animals (Supplementary Fig. 2c, d). Mice transplanted with mDia2-deficient bone marrow cells also exhibited increased lethality (Fig. 1f). When we transplanted mice with bone marrow cells from primary transplantation recipients, mice transplanted with mDia2-deficient bone marrow had markedly shortened survival

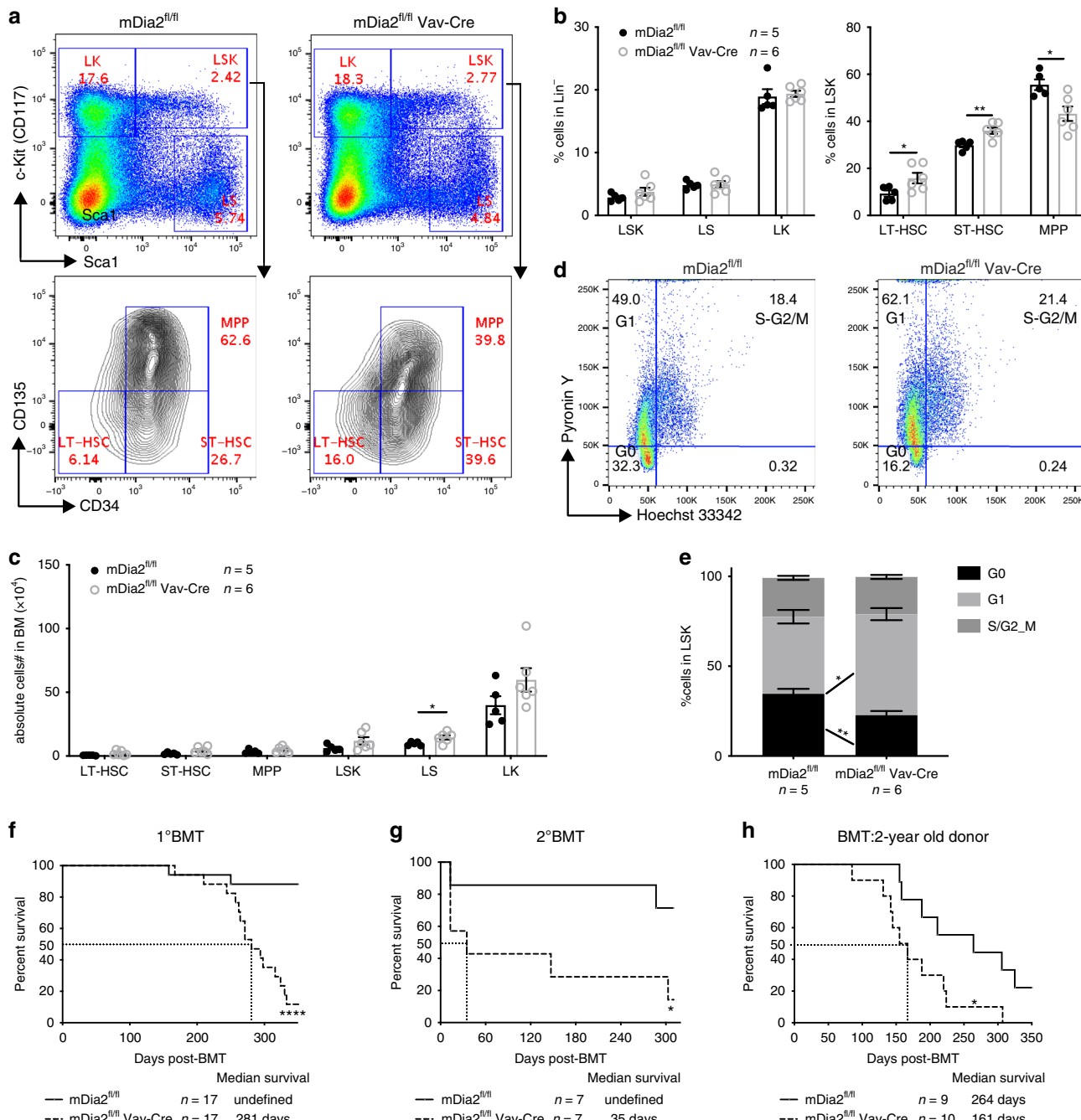

**Fig. 1 mDia2 is required for HSPC engraftment in bone marrow transplantation. a** Representative flow cytometric plots showing gating of different HSPC subpopulations in bone marrow lineage negative cells from wild type mice 4 months after transplantation with $2 \times 10^6$ BMMCs from control or mDia2$^{fl/fl}$ Vav-Cre mice. **b** The percentages of indicated HSPC subpopulations in **a**. **c** The absolute number of indicated HSPCs in **a**. **d** Representative flow cytometric plots of cell cycle profile of LSK cells in **a**. **e** Quantitative analyses of **d**. $n = 5$ mice in mDia2$^{fl/fl}$, $n = 6$ in mDia2$^{fl/fl}$ Vav-Cre for both **b**, **c**, and **e**. **f** Kaplan–Meier survival curve of mice transplanted with $2 \times 10^6$ BMMCs from control or mDia2$^{fl/fl}$ Vav-Cre mice. $n = 17$ mice in each group. **g** Kaplan–Meier survival curve of mice transplanted with $2 \times 10^6$ BMMCs from primary transplanted mice in **e**. $n = 7$ mice in each group. **h** Kaplan–Meier survival curve of wild type mice transplanted with $2 \times 10^6$ BMMCs from 2 years old control or mDia2$^{fl/fl}$ Vav-Cre mice. $n = 9$ mice in mDia2$^{fl/fl}$ group, $n = 10$ in mDia2$^{fl/fl}$ Vav-Cre group. All the error bars represent the SEM of the mean. $*p < 0.05$, $**p < 0.01$, $***p < 0.001$, $****p < 0.0001$. Two-tailed unpaired Student's $t$ test was used to generate the $p$ values. Significance for survival analyses in **f**–**h** was calculated by log-rank (Mantel–Cox) test.

(Fig. 1g), indicating an important role for mDia2 in long-term HSC maintenance. We also performed a BMT assay using donor BMMCs from aged (>2 year) mDia2-deficient mice. The recipient mice showed significantly increased lethality compared to those transplanted with cells from wild type counterparts (Fig. 1h). Taken together, these data demonstrate important functions of mDia2 in maintaining HSPC integrity in BMT.

To further investigate the role of mDia2 under BMT stress conditions, we performed a competitive BMT (cBMT) assay in which an equal number of BMMCs from CD45.2+ mDia2$^{fl/fl}$Vav-Cre mice and CD45.1+ congenic WT mice were transplanted into lethally irradiated wild type CD45.1+ recipient mice (Fig. 2a). Testing of peripheral blood chimerism 5 weeks after transplantation revealed that the absence of mDia2 elicited an

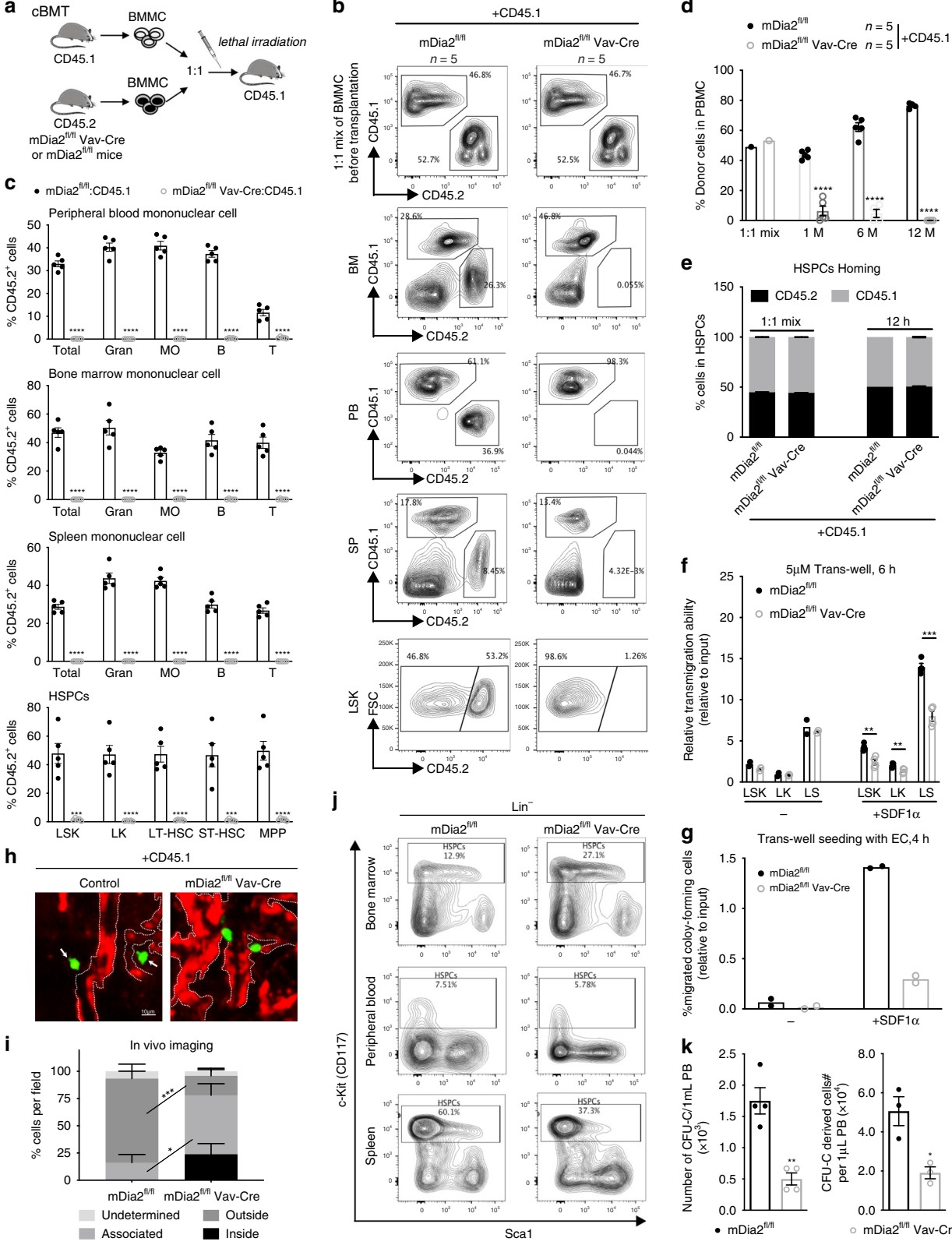

almost complete loss of CD45.2+ cells in the peripheral blood compared to the WT littermate controls (Fig. 2b). Moreover, lineage analyses confirmed a near absence of neutrophils, B, and T cells derived from mDia2-deficient donors (Fig. 2c). Loss of competitive reconstitution of the mDia2-deficient BMMC was also found in the bone marrow and spleen (Fig. 2b, c). Importantly, the absence of mDia2-deficient cells in the LSK,

LK, LT-HSC, ST-HSC, and multipotent progenitor (MPP) populations (Fig. 2b, c) demonstrated that this competitive reconstitution defect was not due to blockage of HSPC differentiation. Engraftment defects were evident from 1 to 12 months after transplantation, indicating that both short-term progenitor and long-term stem cell engraftment were affected by the loss of mDia2 (Fig. 2d). These competitive engraftment

**Fig. 2 Defects in competitive engraftment in mDia2-deficient HSPC. a** Schematic illustration of competitive bone marrow transplantation. **b** Chimerism studies in indicated tissues and bone marrow LSK cells. Representative flow cytometric plots illustrate the percentages of wild type competitive BMMCs (CD45.1+) and the control or mDia2$^{fl/fl}$ Vav-Cre BMMCs (CD45.2+) before and 1.5-month after transplantation. **c** Quantitative analyses of the percentage of donor cells in **b**. Gran: Gr1+ Mac1+ granulocytes; MO: Gr1-Mac1+ monocytes; B: B220+ B cells; T: CD3e+ T cells. **d** Peripheral blood chimerism analyses at the indicated time points. **b–d** n = 5 mice per group. **e** mDia2$^{fl/fl}$ or mDia2$^{fl/fl}$ Vav-Cre bone marrow lineage negative (CD45.2+) HSPCs (2 × 10$^7$ cells) were mixed with equal wild-type CD45.1+ competitive lineage negative cells and transplanted into lethally irradiated wild type mice. Bone marrow chimerism studies were performed at 12 h post-transplantation. Representative data are shown and similar results were repeated twice. **f** Transwell migration assay of HSPCs in the presence or absence of SDF-1α. n = 2 mice in no serum group, n = 4 mice in SDF1a group. **g** Transwell migration assay of indicated lineage-negative cells in the presence of mouse endothelial cells (EC) with or without SDF-1α. n = 2 mice per group. **h** In vivo imaging of fluorescently labeled lineage negative donor cells from the femur of wild-type mice 18 h after competitive transplantation with an equal number (9 × 10$^5$) of non-fluorescently labeled CD45.1+ cells. Red, CD31+ endothelial cells. Green, CD45.2+ donor cells. Dashed white lines outline blood vessels. Repeated twice with similar results. **i** Positions of the indicated donor cells relative to the outlined CD31+ lining sinusoids or arteries were quantified in five fields. N = 39 in mDia2$^{fl/fl}$ group. N = 18 in mDia2$^{fl/fl}$ Vav-Cre group. **j** Representative flow cytometric plots showing percentage of c-Kit+ HSCPs among lineage negative cells after intra-peritoneal injection of G-CSF. **k** Colony forming assay of peripheral blood in **j** (n = 4 mice per group). Colony-derived cells were quantified on the right (n = 3 mice per group). Error bars represent the SEM of the mean. *p < 0.05, **p < 0.01, ***p < 0.001, ****p < 0.0001. Two-tailed unpaired Student's t test was used to generate the p values.

---

defects of mDia2-deficient HSPCs were also observed in the pIpC-treated mDia2$^{fl/fl}$Mx-Cre mouse model (Supplementary Fig. 2e–g). We further confirmed that this defect in engraftment was due to cell-intrinsic loss of mDia2 in HSPCs, since mDia2$^{fl/fl}$Vav-Cre BMMCs from the primary transplants were also significantly reduced in subsequent competitive transplantations (Supplementary Fig. 2h, i).

**mDia2 is required for HSPC lodgment and mobilization.** Defects in the competitive transplantation assay involving mDia2 knockout bone marrow could also be due to compromised short-term HSPC homing. To test this possibility, we first analyzed the expression levels of CXCR4, which senses SDF-1 and is critical for HSPC homing and engraftment[31], in mDia2-deficient HSPCs. In comparing control and mDia2-deficient HSPCs, we observed no significant changes in CXCR4, cell death or ex vivo proliferation profiles (Supplementary Fig. 3a–c). We next performed an in vivo homing assay and found that mDia2-deficient HSPCs migrated to the bone marrow 12, 24, and 48 h after BMT to a degree that was comparable to the WT controls (Fig. 2e, Supplementary Fig. 3d–f). These data indicate that the initial HSPC bone marrow localization is not affected by the loss of mDia2 in vivo. However, this short-term homing assay cannot detect defects in trans-endothelial migration, a key process for HSPC lodgment[1,32]. Indeed, loss of mDia2 was associated with a reduction in HSPC trans-well migration capacity in vitro in response to SDF1α in the presence or absence of endothelial cells (EC) (Fig. 2f, g). To directly determine trans-endothelial migration in vivo, we performed an imaging assay to visualize fluorophore-labeled donor HSPCs and sinusoidal vessels in whole-mount long bones from recipient animals 18 h after competitive transplantation. We found that many donor HSPCs from mDia2$^{fl/fl}$Vav-Cre mice were either trapped inside the vasculature or associated with the vessels, whereas the WT counterparts were readily detected outside of the vessels in the presence of CD45.1+ congenic competitors (Fig. 2h, i, Supplementary Fig. 4a). To confirm this finding under more physiologically relevant conditions, we performed a transplantation assay in non-irradiated recipient mice. The recipient mice exhibited the same trans-endothelial migration defects in mDia2-deficient HSPCs in the bone marrow and spleen (Supplementary Fig. 4b, c). We also performed a competitive transplantation assay involving injection of donor BMMCs directly into the bone marrow through an intrafemoral route, which partially circumvented the engraftment defect in mDia2-deficient HSPCs (Supplementary Fig. 4d).

Lack of trans-endothelial migration could also lead to the reduction in G-CSF-induced HSPC mobilization out of the niche[33]. To test this, we treated mDia2$^{fl/fl}$Vav-Cre mice with G-CSF to induce HSPC mobilization. We found that mDia2$^{fl/fl}$Vav-Cre mice had significantly fewer HSPCs in circulation compared to control mice (Fig. 2j, k). Taken together, these results reveal that mDia2 is required for both HSPC engraftment and lodging in the bone marrow niche.

**mDia2-SRF signaling is involved in HSPC engraftment.** mDia formin proteins are critical for linear actin polymerization[8]. We indeed observed significant downregulation in F-actin filaments in c-Kit+ HSPCs from mDia2-deficient mice (Fig. 3a–c). Downregulation of F-actin was also associated with the changes in cell polarity in which the bipolar distribution of F-actin and alpha-tubulin that is commonly observed in the wild type HSPCs[34–36] was absent in mDia2-deficient cells (Fig. 3a, b). In addition, the prevalence of F-actin protrusions (Fig. 3a arrows) was significantly diminished in mDia2-deficient HSPCs (Fig. 3d).

Dysregulation of the actin cytoskeleton associated with loss of mDia2 was reported to affect cell functions through inactivation of the MAL-SRF pathway in cell-based assays[8,37,38]. In this respect, mDia formin-mediated actin polymerization consumes G-actin monomers from MAL (also known as MRTFα and MKL1). The released MAL rapidly accumulates in the nucleus where it dimerizes with serum response factor (SRF) and drives SRF-dependent gene expression[39]. Supporting this hypothesis, recent genetic studies demonstrated that both SRF and MAL are required for HSPC colonization of the bone marrow and chemotaxis in response to SDF-1α[40]. To determine whether the actin–MAL–SRF axis is impaired by loss of mDia2 in vivo, we stimulated the c-Kit+ HSPCs with serum, which induced MAL nuclear localization in the wide-type cells. However, in HSPCs from mDia2$^{fl/fl}$Vav-Cre mice, MAL maintained a cytoplasmic localization (Fig. 3e and Supplementary Fig. 5a), indicating that the SRF transcriptional network could be disrupted in mDia2-deficient HSPCs. Therefore, we performed a quantitative RT-PCR analysis of c-Kit+ HSPCs purified from mDia2$^{fl/fl}$Vav-Cre mice. We found that many established SRF target genes, including Acta2, KRT17, Flna, FHL2, Myh9, Actb, and Actg1, were significantly down-regulated relative to the control (Fig. 3f). The level of SRF itself was slightly decreased by the loss of mDia2, consistent with the fact that SRF is also a reciprocal transcriptional target of itself[41].

SRF can influence the expression of various adhesion molecules[40]. Given the role of mDia2 in HSPC trans-endothelial migration, we next examined the expression of serval integrins that are critical for HSPC adhesion to EC and trans-endothelial migration. Indeed, the mRNA levels of beta2 subfamily integrins,

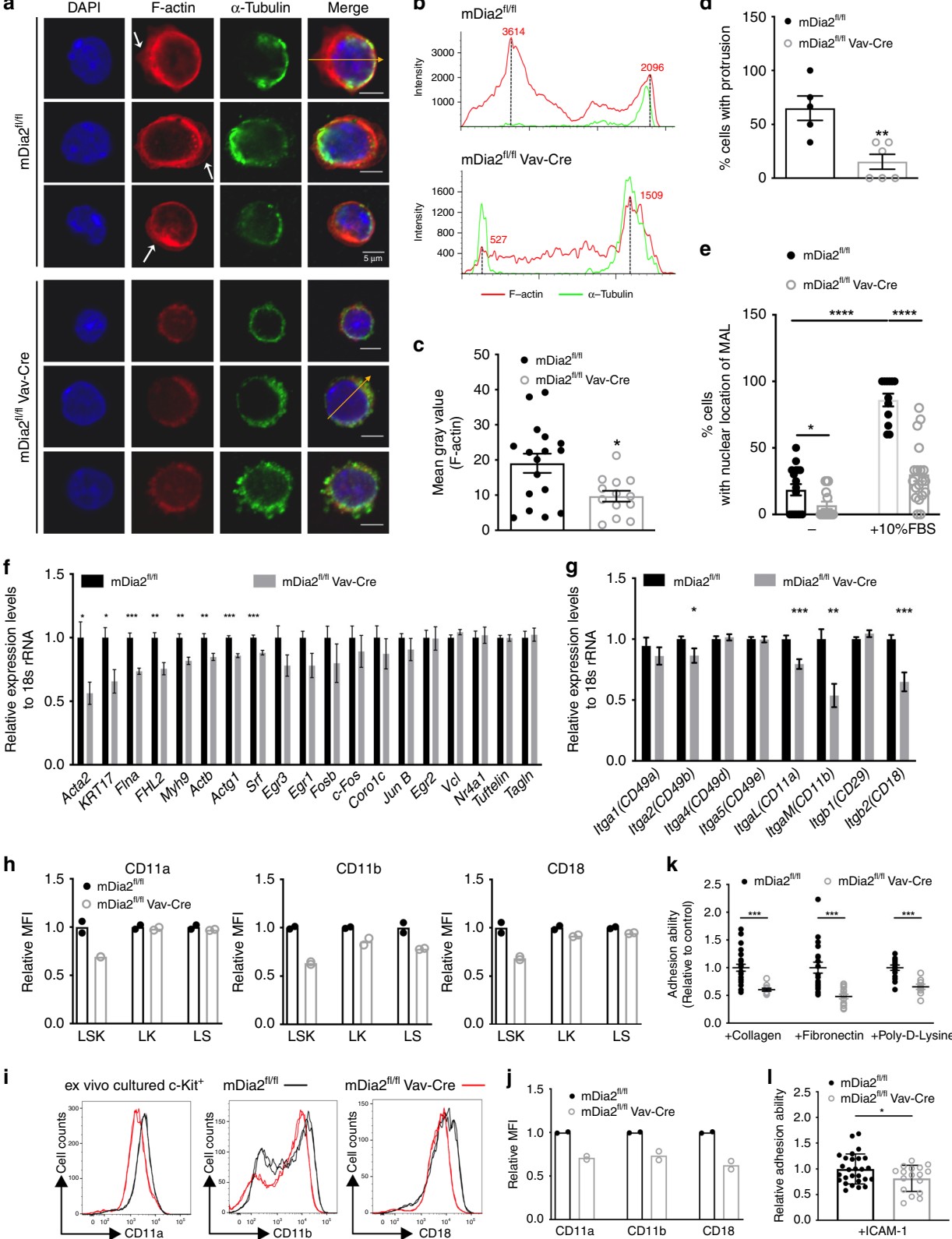

including *ItgaL, ItgaM,* and *Itgb2* that encode CD11a, CD11b, and CD18, respectively, and are known to mediate EC interactions[42], were significantly decreased in bone marrow c-Kit+ HSPCs from mDia2^fl/fl^Vav-Cre mice (Fig. 3g). Serum stimulation dramatically induced the expression of *ItgaM, Itgb2,* and the classic SRF downstream target genes. The decreased expression of these genes was more prominent in mDia2 null cells (Supplementary

Fig. 5b, c). The surface expression of these integrins was also consistently decreased in HSPCs from mDia2^fl/fl^Vav-Cre mice and ex vivo cultured mDia2 null HSPCs (Fig. 3h–j). The expression of beta2 integrins in HSCs was demonstrated in a gene expression database[43] in that the reading peaks in RNA-sequencing and H3K36 trimethylation (an epigenetic mark associated with the activation of RNA polymerase 2) ChIP-sequencing were evident

**Fig. 3 mDia2-SRF signaling is involved in the regulation of HSPC engraftment. a** Immunofluorescence staining of HSPCs with F-actin (red) and α-tubulin. White arrows: F-actin protrusions. Yellow arrows: polarization axis to determine the staining intensity in **b**. The experiments were repeated three times with similar results. **b** F-actin and α-tubulin staining intensity across the yellow arrows from **a**. **c** Quantitative analyses of F-actin staining intensity from **a**. $N = 17$ cells in mDia2$^{fl/fl}$ group. $N = 13$ cells in mDia2$^{fl/fl}$Vav-Cre group. **d** Quantitative analyses of the percentage of cells with F-actin protrusions in **a**. $N = 5$ fields in mDia2$^{fl/fl}$ group. $N = 6$ fields in mDia2$^{fl/fl}$ Vav-Cre group. **e** Quantitative analysis of the percentages of MAL nuclear localization in c-Kit+ HSPCs incubated with or without FBS for 15 min. mDia2$^{fl/fl}$ no FBS: $N = 71$ cells from 17 random fields; mDia2$^{fl/fl}$Vav-Cre no FBS: $N = 80$ cells from 15 random fields; mDia2$^{fl/fl}$ with 10% FBS: $N = 56$ cells from 12 random fields; mDia2$^{fl/fl}$Vav-Cre with 10% FBS: $N = 108$ cells from 19 random fields. Data are from three independent experiments. **f** Quantitative RT-PCR of the indicated SRF target genes in c-Kit+ HSPCs from the indicated mice. **g** Quantitative RT-PCR of the indicated integrins in c-Kit+ HSPCs from the indicated mice. The experiments were performed in triplicate from $n = 5$ mice per group in **f**, **g**. **h** Quantitative analyses of the surface expression level of beta2 integrins using flow cytometry in the indicated HSPC populations. $n = 2$ mice per group. **i** Representative flow cytometry plots showing the expression of beta2 integrins in ex vivo-cultured c-Kit+ HSPCs. **j** Quantitative analyses of **i**. $n = 2$ mice per group. **k** In vitro adhesion assay of the cultured c-kit+ HSPCs from the indicated mice on extracellular matrix protein coated coverslips. Data are presented as adhesion ability relative to the control cells. mDia2$^{fl/fl}$, Collagen: 25 fields, Fibronectin: 20 fields, poly-D-lysine: 15 fields. mDia2$^{fl/fl}$ Vav-Cre, Collagen: 10 fields, Fibronectin: 14 fields, poly-D-lysine: 9 fields. **l** In vitro adhesion assay of indicated c-kit+ HSPCs cultured on ICAM-1-coated coverslips. Error bars represent the SEM of the mean. *$p < 0.05$, **$p < 0.01$, ***$p < 0.001$, ****$p < 0.0001$. Two-tailed unpaired Student's *t* test was used to generate the *p* values.

for both *ItgaM* and *Itgb2*. In contrast, transcription of proximal *Itgad* loci near *ItgaM* was barely detectable (Supplementary Fig. 5d–g). To determine whether downregulation of these integrins affects HSPC adhesion, we performed an adhesion assay using ex vivo cultured HSPCs and revealed compromised adhesion of mDia2-deficient HSPCs to several bone marrow extracellular matrix components (Fig. 3k) that have been shown to bind to beta2 integrins in hematopoietic cells[44,45]. In coverslips coated with ICAM-1, a main physiological substrates of beta2 integrins[46], mDia2-deficient HSPCs also showed adhesion defects (Fig. 3l). Collectively, these data indicate that EC binding and trans-endothelial migration defects in mDia2-deficient HSPCs are likely due to altered expressions of beta2 integrins mediated by the MAL–SRF pathway.

**Beta2 subfamily integrins are targets of SRF.** Homodimeric SRF binds to a consensus cis-element, known as the CArG box (CC (A/T)6GG) or serum response element (SRE), on its target genes[47,48]. To determine whether beta2 integrins are bone fide transcriptional targets of SRF, we used a SRF chromatin-binding database[49] with a transcription factor-binding site prediction tool (TFBIND)[50] and identified intronic binding of SRF on both *ItgaM* and *Itgb2* genomic loci (Fig. 4a), but not on *ItgaL*. Therefore, we focused subsequent studies on *ItgaM* and *Itgb2*. SRF presents as several isoforms with exon 3, 4, or 5 deleted singly or in combination (Fig. 4b). In addition to the full-length SRF (SRF-FL), the isoform with deletion of exon 5 (SRFΔ5) was also highly expressed in sorted LK or LSK cells (Fig. 4b). We cloned the intronic region of either *ItgaM* or *Itgb2* that contains the predicted SRE and inserted it ahead of the luciferase gene (Fig. 4c). As expected, full-length SRF strongly and dose-dependently induced luciferase gene expression when co-transfected with luciferase reporters containing either the *ItgaM* or *Itgb2* intronic region in 293T cells (Fig. 4d, e). Notably, the orientation of these SRE elements exhibited marked differences in the ability to enhance luciferase activities, indicating a possible complex three-dimensional chromatin organization in vivo (Fig. 4c). SRFΔ5-activated luciferase activity to a lesser degree compared to SRF-FL, but the level was comparable to the constitutively active SRF-VP16 in 293T cells (Supplementary Fig. 6a). To further validate these SREs, we mutated the consensus C/G nucleotides (AA mutant), the adjacent T to C (TC mutant), or these two sites combined (AACT mutant) in the intronic SRE of *ItgaM*. Each change was associated with dramatic reductions in SRF-induced luciferase activities (Fig. 4f). The SRE on *Itgb2* intron is non-classic. Deletion of this site blocked the luciferase

activity (Fig. 4g). Since C/G is critical for the consensus sequence, we mutated C and G in this non-classic SRE site to produce a G/C to T mutation. As expected, this mutation also abolished luciferase activity (Fig. 4g). Using chromatin immunoprecipitation (ChIP) assays of c-Kit+ HSPCs, we further confirmed SRF and MAL binding to these SREs, as well as SRFs established target *Acta2*, which was abolished by the loss of mDia2 (Fig. 4h and Supplementary Fig. 6b). Next, we depleted the intronic SRE of *ItgaM* in HSPCs using a CRISPR-Cas9 system, which also significantly affected CD11b expression (Fig. 4i and Supplementary Fig. 6c–g). Overall, these results reveal functional SRF cis-elements in beta2 integrins.

We next asked whether over-expression of SRF in HSPCs could promote the expression of beta2 integrins in vitro. Indeed, SRF-FL-, SRFΔ5- or SRF-VP16-transduced c-Kit+ HSPCs showed higher expression levels of CD11b and CD18 (Fig. 4j), as well as CD11a (Supplementary Fig. 7a), than the control vector. Notably, SRF-VP16 exhibited more potent activity than SRF-FL and SRFΔ5. However, SRF-VP16 induced significant apoptosis and inhibition of c-Kit+Sca1+ population (Supplementary Fig. 7b, c). This indicates that SRF activity is tightly regulated in that hyperactivation of SRF compromises HSPC survival and self-renewal.

**SRF restores engraftment defects in mDia2-deficient HSPCs.** To establish a proof-of-concept, we next analyzed whether SRF overexpression could rescue the engraftment defects in vivo in mDia2-deficient HSPCs. We first showed that overexpression of SRF-FL or SRFΔ5 rescued protein expression of beta2 integrins in mDia2$^{fl/fl}$ Vav-Cre HSPCs in vitro (Fig. 5a and Supplementary Fig. 7d). A constitutively active mDia2 mutant with a deletion of the C-terminal DAD domain (ΔDAD) also strongly rescued beta2 integrin expression even with a low transduction efficacy that was partially due to the insertion of a large open-reading frame (ORF) into the vector. In contrast, an N-terminal GBD/DID mutant lacking actin polymerization domains had no rescue effect, further emphasizing the significance of actin polymerization activity in mDia2-mediated expression of beta2 integrins. We also examined multiple mRNA transcripts in mDia2-deficient HSPCs after SRF overexpression by quantitative RT-PCR. Both SRF-FL and Δ5 isoforms normalized the mRNA levels of beta2 integrins and several known SRF target genes to wild type levels (Fig. 5b and Supplementary Fig. 7e). In addition, SRF overexpression rescued the compromised adhesion ability of mDia2-deficient HSPCs in vitro (Fig. 5c).

To determine the rescue effects of SRF in vivo, we used competitive transplantation assays in which SRF-FL, SRFΔ5, or

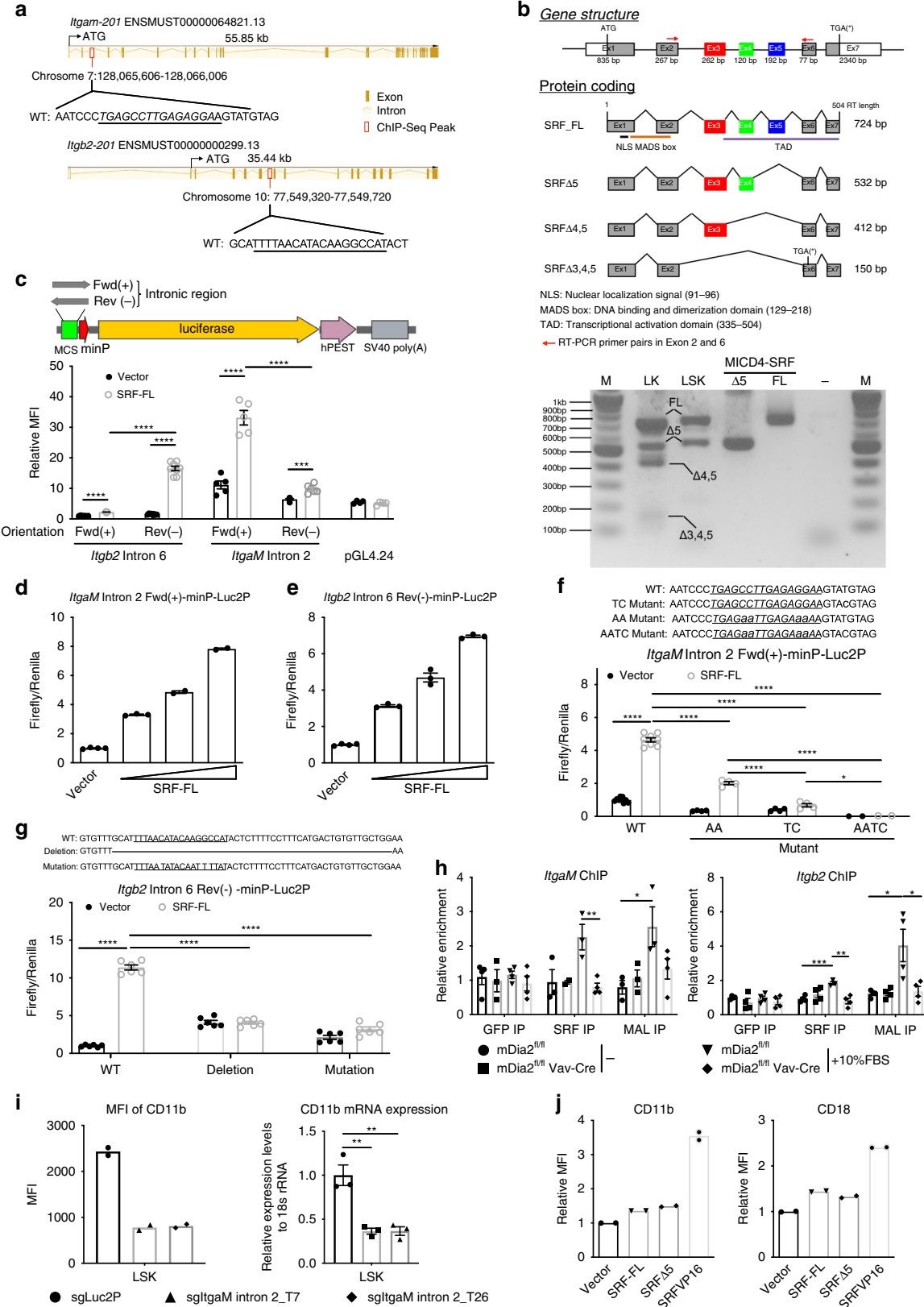

mDia2 mutants were transfected into CD45.2+ c-Kit+ HSPCs cells from mDia2^fl/fl Vav-Cre mice. Equal numbers of these cells were transplanted with an equal number of empty vector-transduced wild type CD45.1+ HSPCs into lethally irradiated CD45.1+ recipient mice. Chimeras in peripheral blood could be observed by flow cytometry for up to 6 months post-transplantation (Fig. 5d). Re-expression of WT mDia2 or the constitutively active mDia2 mutant (ΔDAD) completely rescued the engraftment defects in competitive transplantation of mDia2-deficient HSPCs in vivo, even with low transduction efficiencies (Fig. 5e and Supplementary Fig. 7f). Overexpression of both SRF-FL and SRFΔ5 also significantly rescued the

**Fig. 4 Members of beta2 subfamily integrin are targets of SRF. a** Schematic map of SRF ChIP-seq peak in the genomic regions of mouse *ItgaM* and *Itgb2*. Putative SREs are underlined. **b** Full-length and variants of SRF in murine hematopoietic cells (upper panel). SRF variants were amplified by conventional RT-PCR from the indicated cells or constructs and assessed by gel electrophoresis (lower panel). The experiments were repeated three times with similar results. **c** SRF enhancement of luciferase activities in reporter constructs harboring the intronic regions of *Itgb2* or *ItgaM* is orientation dependent. 293T cells transiently expressing the indicated luciferase reporters were co-transfected with empty vectors or an SRF-FL-expressing construct in the presence of a Renilla luciferase expression vector (pRL-TK). Firefly and Renilla luciferase activities were measured after 24 h. **d** Up-regulation of luciferase gene expression driven by *ItgaM* intronic sequence with increased expression of SRF-FL in 293T cells. **e** Same as **d** except *Itgb2* intronic region was tested. **f** The same luciferase activity assays as in **c** were performed using 293T cells co-transfected with either the control vector or SRF-FL plus luciferase reporters containing either intact (WT) or mutated SRE (Mutant). WT: $n = 9$ with vector, $n = 8$ with SRF-FL, AA and TC: $n = 4$, AATC: $n = 2$. **g** Same luciferase assays as **f** except *Itgb2* mutants were tested. $n = 6$ in each group. **h** Chromatin immunoprecipitation assays with indicated antibodies in untreated or 10% FBS-stimulated c-Kit+ HSPCs from indicated mice. DNA fragments containing the SREs of *ItgaM* (left) and *Itgb2* (right) were amplified and detected by real-time PCR in quadruplicate. Data are shown as the quantitation of signal relative to input chromatin. **i** Quantitative analyses of CD11b protein (left, $n = 2$ mice per group) and mRNA (right, triplicate) expressions in LSK cells from Cas9-GFP transgenic mice transduced with indicated sgRNAs. **j** Quantitative analyses of surface expression of CD11b and CD18 detected by flow cytometry in wild-type c-kit+ HSPCs transduced with indicated genes. $n = 2$ mice per group. Error bars represent the SEM of the mean. $*p < 0.05$, $**p < 0.01$, $***p < 0.001$, $****p < 0.0001$. Two-tailed unpaired Student's *t* test was used to generate the *p* values.

competitive engraftment defects in mDia2-deficient HSPCs (Fig. 5e). The rescue effects persisted over the long term and in multi-lineages (Fig. 5e, f). More importantly, CD45.2+ LT-HSC and LSK cells were readily detected at 6 months post-transplantation in these rescued mice (Fig. 5g). To further demonstrate the significance of SRF in mediating the functions of mDia2 in HSPC engraftment, we performed a serial transplantation assay in which various mDia2 or SRF constructs were transduced into mDia2-deficient c-Kit+ HSPCs before use in serial transplants. As expected, mDia2 WT, ΔDAD, and SRF-FL, SRF-Δ5 significantly extended the survival of the recipients with a secondary bone marrow transplant (Fig. 5h). Taken together, these results demonstrate that mDia2 regulates HSPCs in vivo mainly through the SRF signaling.

**Beta2 integrins mediate mDia2 function in HSPC engraftment.** CD18 interacts with CD11a and CD11b to form LFA1 and Mac1, respectively, which are critical for hematopoietic cell migration and adhesion[51,52]. However, the roles of these complexes in HSPC engraftment are unclear. To study hematopoietic-specific loss of function of beta2 integrins in vivo, we constructed hematopoietic-specific knockout animal models of CD11b or CD18 deficiency using CRISPR-Cas9 (Fig. 6a). Before the sgRNAs were introduced into HSPCs, their targeting efficacy was measured through a rapid and quantitative luciferase activity-based assay. Specifically, the sgRNA target sites on CD11b or CD18 were cloned in-frame within the region of the *luc2P* gene encoding the N-terminus. The clones were then co-transfected with sgRNA expression vectors into 293T cells stably expressing CRISPR-associated protein 9 (Cas9). The effective Cas9/sgRNA complex is expected to bind to target DNA sequences causing out-of-frame indels that abrogate the luciferase activity (Fig. 6a). Using this approach, we obtained two sgRNAs that separately targeted *ItgaM* and *Itgb2* with over 90% efficiency (Fig. 6b). We next transduced sgRNA lentiviral particles into c-Kit+ HSPCs from mice that constitutively and ubiquitously express Cas9 and EGFP and transplanted these cells into lethally irradiated CD45.1+ recipient mice (Fig. 6a). We archived nearly 90% transduction efficiency as verified in peripheral blood mononuclear cells from these mice (Fig. 6c). As a consequence, cell surface expression of CD18 and CD11b was almost completely abolished in Gr1+ granulocytes and LSK cells (Fig. 6d, e). Thus, we obtained mice with either separate *ItgaM* or *Itgb2* deficiency or deficiency of both genes in hematopoietic cells. In competitive transplantation assays, knockout of CD11b or CD18 individually or together was associated with failure of competitive engraftment (Fig. 6f). We also performed secondary BMT assays to test the engraftment and self-renewal capacity of HSCs after loss of beta2

integrins. Recipient mice transplanted with HSPCs from combined CD11b and CD18 knockout died rapidly after transplantation compared to mice transplanted with HSPCs from mice lacking only CD18 or control HSPCs (Fig. 6g). Collectively, our data suggest that beta2 integrins are indispensable for HSPC engraftment and long-term self-renewal of HSCs.

We next tested whether mDia2 deficiency-induced defects in HSPC engraftment could be rescued by ectopic expression of beta2 integrins. We found that ectopic expression of *ItgaM*, but not *Itgb2*, partially restored the competitive engraftment defect in mDia2[fl/fl] Vav-Cre c-kit+ HSPCs (Fig. 6h, i). Co-ectopic expression of *ItgaM* and *Itgb2* reverted the rescue effect of *ItgaM* (Fig. 6i). These data reveal that *ItgaM*, but not *Itgb2*, is the limiting factor on the mDia2-SRF signaling cascade involved in HSPC engraftment. They also indicate that CD18 levels must be precisely regulated to avoid adverse effects on HSPCs mediated by CD18 overexpression.

## Discussion

Successful bone marrow engraftment requires HSPC homing to the bone marrow vasculature, trans-endothelial migration, and lodging to the bone marrow niche. Research over the past few decades has revealed proteins that play key roles in HSPC homing and engraftment including selectins, ICAM-1, VCAM-1 on the EC[53,54], beta1 subfamily integrins, and CXCR4 on HSPCs[2,55,56]. However, the mechanisms involved in trans-endothelial migration of HSPCs were unclear since loss of function studies focusing on these proteins showed HSPC defects mainly in short-term homing or long-term engraftment processes. Although vascular endothelial cadherin was indicated to play a role in HSPC migration across bone marrow endothelium[57], proteins expressed on HSPCs in the trans-endothelial process are rarely studied. In this study, we reveal a role for the formin protein mDia2 in the engraftment of HSPCs during transplantation. The mDia2 hematopoietic-specific knockout mouse model overcomes the embryonic lethality of whole body mDia2 knockout and provided us an opportunity to determine the role of mDia2 in HSPCs in adults[25,28]. We found that loss of mDia2 in adult mice significantly compromised competitive HSPC engraftment and long-term HSC repopulation capacities. Surprisingly, HSPC localization to the bone marrow vasculature in these mice remained intact. Through imaging analyses, we revealed that mDia2 is critical for the trans-endothelial migration of HSPCs.

mDia2 is also critical for HSPC adhesion to the bone marrow niche. Here we found decreased adhesion of mDia2 knockout HSPCs to extracellular matrix proteins in vitro. Loss of mDia2 was associated with a slight expansion of HSPCs in the bone

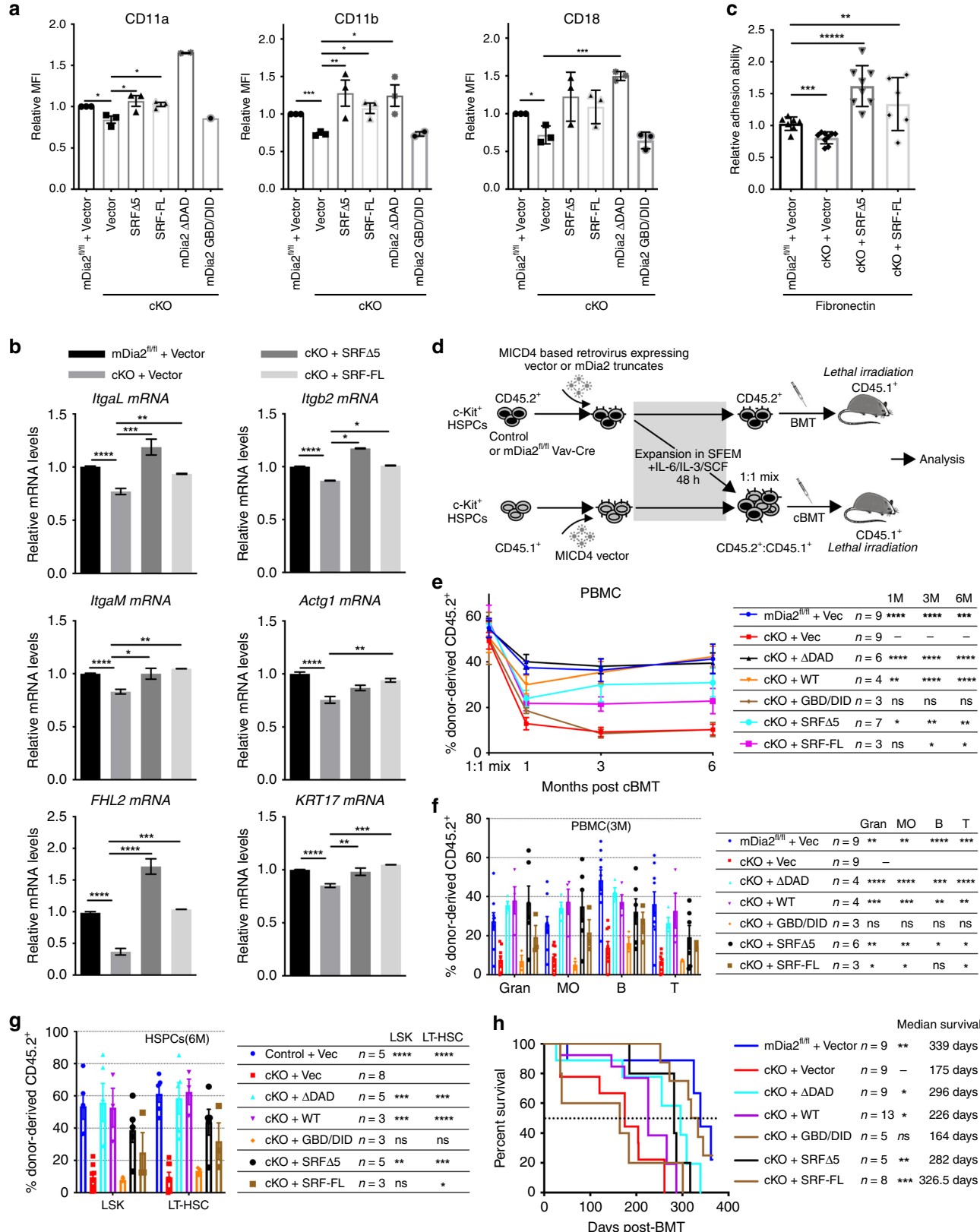

marrow along with loss of HSC quiescence in the early stages after transplantation. This result is consistent with previous studies that ICAM-1 deficiency in the bone marrow niche impairs HSC quiescence[58]. The defects in HSC quiescence defects continued through the late stages of transplantation that eventually led to exhaustion and reduction in HSPC populations.

We further demonstrated that mDia2 functions through the MAL–SRF signaling axis to regulate HSPC engraftment. Overexpression of SRF, including a naturally occurring SRF isoform with a deletion of exon 5, rescued the engraftment and long-term HSC reconstitution defects in mDia2-deficient HSPCs. These results are consistent with a previous study on the role of SRF in

**Fig. 5 SRF restores engraftment defects in mDia2-deficient HSPCs. a** Quantitative analysis of the flow cytometry detected surface expression levels of beta2 integrins in c-Kit+ HSPCs from mDia2[fl/fl] or mDia2[fl/fl]Vav-Cre (cKO) mice transduced with indicated genes. $N = 3$ mice in each group, except mDia2 ΔDAD and mDia2 GBD/DID groups. **b** Restoration of the mRNA expression of beta2 integrins and indicated SRF downstream genes by overexpression of SRF in c-Kit+ cells from mDia2[fl/fl]Vav-Cre (cKO) mice. The experiments were performed in triplicate. $N = 3$ mice in each group. **c** Viral transduced c-kit+ HSPC from mDia2[fl/fl] or mDia2[fl/fl]Vav-Cre (cKO) mice were applied for in vitro adhesion assay with fibronectin precoated coverslip for 1 h at 37 °C. Adherent cells were randomly scored from $N = 7$ fields in mDia2[fl/fl]+Vector, $N = 9$ fields in cKO+Vector, $N = 8$ fields in cKO+SRFΔ5, and $N = 6$ fields in cKO+SRF-FL. Results are expressed as the adhesion ability relative to control cells transduced with vector. **d** Schematic diagram showing bone marrow transplantation experiments in **e**–**g**. **e** Chimerism studies of peripheral blood mononuclear cells derived from c-Kit+ cells that were transduced with the indicated constructs after competitive transplantation with CD45.1+ cells at indicated timepoints. **f** Quantification of multi-lineage reconstitutions in the peripheral blood from **e** at 3-month. **g** LSK and long-term HSC engraftment analyzed in bone marrow from 6-month-old transplanted mice shown in **e**. **h** Kaplan–Meier survival curve of indicated groups of mice after secondary transplantation illustrated in **d**. Error bars represent the SEM of the mean. $p$ values compared to cKO+Vec group from **g**–**j** are presented on the right table. $*p < 0.05$, $**p < 0.01$, $***p < 0.001$, $****p < 0.0001$. Two-tailed unpaired Student's $t$ test was used to generate the $p$ values. Significance for survival analyses was calculated by log-rank (Mantel–Cox) test.

HSPC engraftment and adhesion to bone marrow niches[40]. In contrast to the mouse model with hematopoietic-specific loss of SRF that showed a significant expansion in HSPCs, we observed no changes in these populations in mDia2-deficient mice under steady-state conditions. Instead, the HSPC defects were mainly observed during BMT. This feature could be due to a compensatory effect of mDia1. Furthermore, given that SRF regulates multiple target genes, loss of SRF could induce HSPC phenotypes under steady-state conditions that were caused by other genes targeted by SRF.

In this respect, the role of mDia2 in HSPC function is apparently more specific than that of SRF in that the mDia2–MAL–SRF axis is critical for trans-endothelial migration that is part of the complex journey taken by HSPCs during homing and engraftment. This specificity is also evident in that only a subset of SRF downstream targets is affected by the loss of mDia2 in HSPCs. Among these down-regulated genes, we discovered that beta2 subfamily integrins, including *ItgaM* and *Itgb2*, are direct SRF target genes. Beta2 integrins are mainly expressed in granulocytic cells. Accumulating evidence demonstrates that these integrins are also expressed on HSPCs. For example, a large proportion of the side population of HSCs co-express low levels of CD11b[59] and nearly half of phenotypic mouse HSCs (defined as CD150+CD34−LSK) are CD11a positive[60]. Furthermore, long non-coding RNAs specifically expressed in HSCs (LncHSCs) co-occupy *Itgb2* promoter regions with the E2A transcription factor to facilitate their expression[61]. These integrins are also functionally important for HSPCs. Specifically, Mac1 (CD11b/CD18 complex) mediates the adhesion of hematopoietic progenitor cells to stromal cell elements[62]. Meanwhile, ITGAM was reported to colocalize with GPI-80, which is critical for human HSC in vitro expansion and engraftment[63].

We further demonstrated the significance of beta2 integrins in regulating the mDia2–SRF pathway by showing that *ItgaM* overexpression can rescue defects in competitive transplantation of mDia2 null HSPCs. However, *Itgb2* overexpression failed to rescue this phenotype. We thus hypothesize that CD18 levels must be precisely regulated in that retroviral-mediated overexpression could be toxic to HSPCs. Consistent with this hypothesis, overexpression of both *ItgaM* and *Itgb2* also failed to rescue the defect. The complexity of CD18 activity was highlighted by earlier in vivo studies involving a hypomorphic CD18 mutant mouse model that revealed an expansion of HSPCs in the bone marrow in competitive transplantation assays[64]. In the present study we found that CD18-deficient HSPCs, generated through a CRISPR-Cas9 approach, lost their competitive engraftment capacities in a manner similar to that seen for mDia2 deficiency. This discrepancy indicates that hypomorphic CD18 could have potential gain-of-function properties.

SRF recognizes a SRE on its target genes that is commonly known as the CArG element (CC(A/T)6GG)[48]. SREs are typically classified into two categories, consensus CArG and CArG-like[48]. The SRE on *ItgaM* is consistent with the CArG-like element. Although the SRE on *Itgb2* is non-classical, recent studies using SRF ChIP coupled with next generation sequencing (ChIP-seq) revealed SREs that lack CC(A/T)6GG consensus sequence in many SRF-regulated genes[65]. These SREs on *ItgaM* and *Itgab2* are also orientation-dependent as demonstrated by the luciferase assay in this study that suggested a complex chromosome organization for SRF binding and regulation via these elements. This was further indicated by an incidental finding that a thymidine nucleoside 3′ downstream of the *ItgaM*'s SRE is critical for the recognition of SRF. However, it is also possible that this site affects binding of another factor that could be required for SRF to be functional.

We previously reported that loss of mDia1 compromised endocytosis and led to dysregulation of CD11b on neutrophil cell membranes[22]. In contrast, findings in the current study demonstrated that expression of beta2 subfamily integrins, including CD11b, is downregulated both at the mRNA and cell membrane protein level in the HSPCs from mDia2-deficient mice. Notably, these cell type-specific functions are also found in SRF[66]. In this respect, previous studies in neutrophils revealed that loss of SRF did not significantly affect the mRNA level of many integrins. Instead, cell membrane expression levels of beta2 integrins were upregulated in SRF null neutrophils due to the lack of internalization and recycling of these integrins[67]. The findings of the present study are consistent with the finding that upregulated CD11b in mDia1 null neutrophils is due to compromised endocytosis activity and suggests a possible connection of mDia1 with SRF in neutrophils as well. Moreover, these studies indicate that, similar to SRF, mDia formins also play cell type-specific and content-specific roles in that in HSPCs, mDia mainly influences the transcriptional activity of SRF to affect beta2 integrin expression, whereas in neutrophils, mDia mainly functions to regulate integrin endocytosis to affect their cell membrane expression.

## Methods

**Mice.** Genetically modified tissue-specific mDia2 knockout mice with a C57/BL6 background were described previously[25,26]. Congenic mice carrying CD45.1 antigen were purchased from Charles River (B6-LY-5.2/Cr, strain code: 564). C57/BL6 WT mice, CAG-Cas9 transgenic mice (stock #026179), Mx-Cre mice (stock #003556), and Vav-Cre mice (stock #008610) were purchased from the Jackson Laboratory. All the experiments involving animals were conducted in accordance with the Guide for the Care and Use of Laboratory Animals and were approved by the Institutional Animal Care and Use Committee at Northwestern University.

**Expression constructs.** The MSCV-IRES-hCD4 (MICD4) construct was described previously. All inserts were sub-cloned 5′ to the IRES[24,68]. Murine SRF ORF was PCR amplified using cDNA reverse transcribed from total RNA of mouse bone marrow lineage negative cells. Two isoforms including SRF full length (SRF-FL) and Exon 5-deficient truncation (SRFΔ5) were obtained. The expression construct

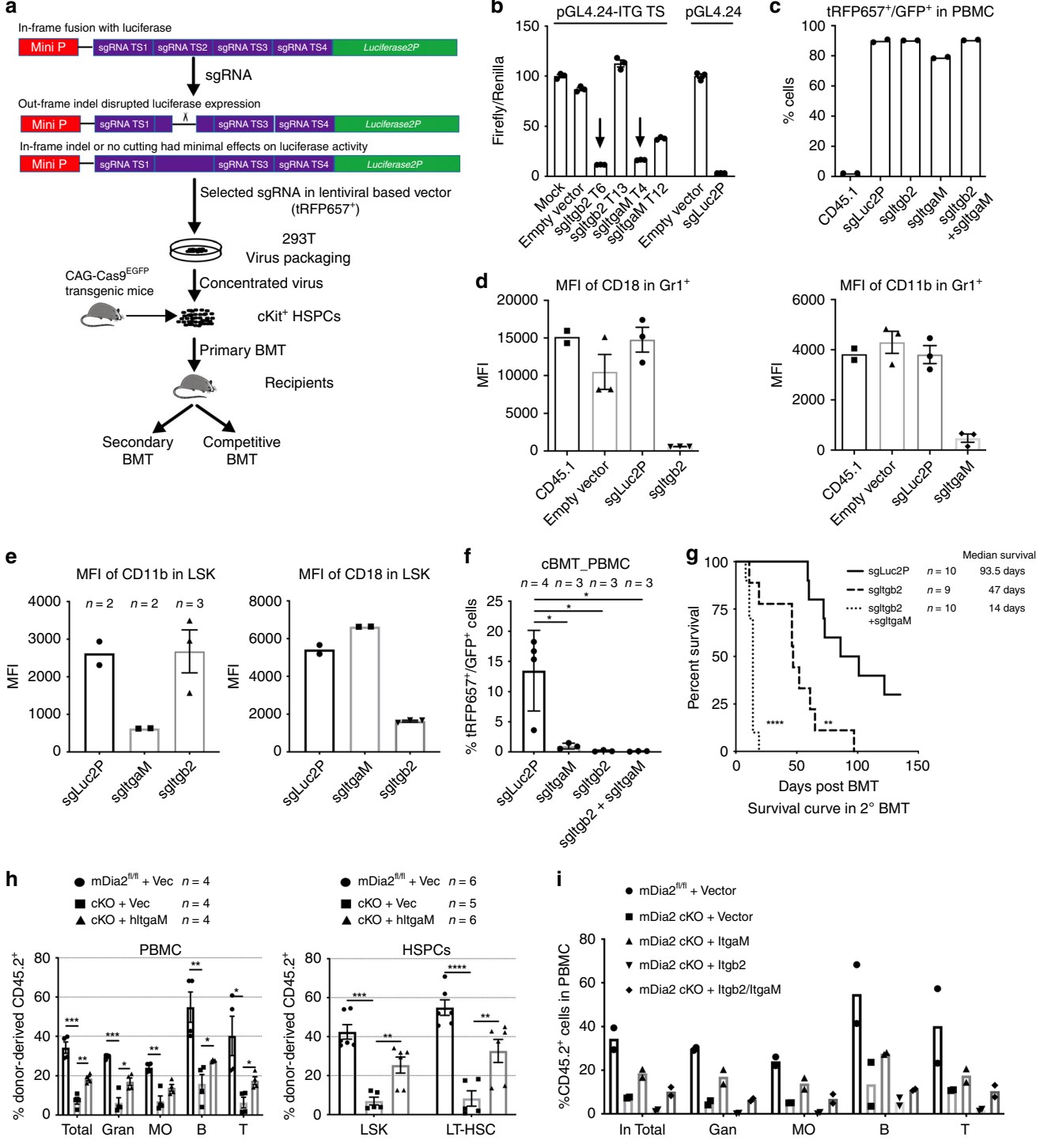

carrying a HA-tagged, constitutively active form of SRF was kindly provided by Dr. Naren Ramanan (Centre for Neuroscience Indian Institute of Science, India), and was PCR amplified and sub-cloned into the MICD4 vector. The MICD4-Flag-mDia2 WT expression construct was described previously[15]. A flag-tagged constitutively active form of mDia2 with a DAD domain deletion (mDia2 ΔDAD) (1–1031aa) and the HA-tagged mDia2 GBD/DID mutant (1–531 aa) were generated by inserting the PCR-amplified mutants into the MICD4 vector. The mouse Itgb2 expression construct pCMV-Itgb2 (Catalog #MG50359-UT) was purchased from Sino Biological Inc. (Beijing, China). The human ITGAM expression plasmid was a gift from Timothy Springer (Addgene plasmid #8631). Both Itgb2 and ITGAM ORFs were PCR amplified and sub-cloned into the MICD4 vector. The primer sequences are listed in Supplementary Table 1.

**Flow cytometric assays**. The preparation of single cell suspensions from mouse bone marrow and spleen tissue was described previously[25]. In brief, bone marrow

cells from femur and tibia were flushed out using a syringe with 30.5 G needle. The spleen was dispersed by homogenization using the frosted ends of the slides. Single cell suspensions were obtained by pipetting and passing through 40 µm cell strainer. To prepare cell suspensions from peripheral blood, 5–8 drops of tail blood were collected directly into 1 ml FACS buffer (1× PBS containing 0.5% BSA and 2 mM EDTA). After centrifugation at $6000 \times g$ for 5 min, the cell pellets were lysed by incubation in 1× RBC lysis buffer (Catalog #00-4333-57, Thermo Fisher Scientific) for 5 min. The nucleated cells (white blood cells) were recovered and washed by centrifugation at $6000 \times g$ for 5 min before the cells were stained with PE-CD11b, PE-Cy7-Gr1, Pacific Blue-B220, APC-eFluor780-CD3e, or APC/Fire 750-CD3e, FITC-CD45.2 or V500-CD45.2, and APC-CD45.1 in 100 µl FACS buffer for the analyses of lineage distributions.

Lineage-negative HSPCs were isolated from the tibia and femoral marrow compartments by depletion of lineage-positive cells using a Lineage Cell Depletion Kit (Catalog # 559971 BD Pharmingen) according to the manufacture's protocol.

**Fig. 6 Beta2 integrins mediate mDia2's function in HSPC engraftment. a** Experimental outline for the measurement of sgRNA genome editing activities by a luciferase reporter assay and subsequent editing specifically in hematopoietic cells from Cas9[EGFP] transgenic mice through bone marrow transplantation. **b** Methodological validation of luciferase-based genome editing in Cas9 expressing 293T cells. Representative data performed in triplicate were shown. The experiments were repeated twice with similar results. **c** Quantitative analyses of the transduction efficiencies of the indicated sgRNAs in vivo in the hematopoietic cells by flow cytometry one month after transplantation. $n = 2$ mice in each group. **d** Evaluation of genome editing efficacies of sgRNAs in peripheral blood Gr1+ granulocytes. $n = 3$ mice per group except $n = 2$ mice in CD45.1 as negative control. **e** Same as **d** except LSK cells in the bone marrow were analyzed. $n = 2$ mice per group except $n = 3$ in sgItgb2 group. **f** Chimerism analysis in the peripheral blood of the recipient mice following secondary competitive bone marrow transplantation with $2 \times 10^6$ BMMC donor cells transduced with the indicated sgRNAs and equal number of CD45.1+ wild type BMMC competitors. $n = 3$ mice per group except $n = 4$ in sgLuc2P group. **g** Kaplan–Meier survival curve of indicated mice after secondary transplantation with $3.5 \times 10^6$ BMMCs from primary transplant recipients from **c**. $n = 10$ mice per group except $n = 9$ in sgItgb2 group. **h** c-Kit+ HSPCs transduced as indicated were applied for competitive transplantation as described in Fig. 5e. Chimerism studies in the peripheral blood at 4 months after transplantation were shown on the left panel ($n = 4$ mice in each group), and the long-term HSC engraftments in the bone marrow were on the right panel ($n = 6$ mice per group except $n = 5$ in cKO+Vec group). **i** Chimerism analyses of multi-lineage reconstitutions in peripheral blood mononuclear cells derived from indicated transduced c-Kit+ cells after competitive transplantation with CD45.1+ cells. $n = 2$ mice in each group. Error bars represent the SEM of the mean. $*p < 0.05$, $**p < 0.01$, $***p < 0.001$, $****p < 0.0001$. Two-tailed unpaired Student's $t$ test was used to generate the $p$ values. Significance for survival analyses was calculated by log-rank (Mantel–Cox) test.

The cells were then incubated with PE-Sca1(Ly-6A/E), PE-Cy7-CD117(c-Kit), APC-CD135, BV421-CD34, and PerCP-Cy5.5-CD16/CD32 for characterization of LT-HSC/ST-HSC/MPP and GMP/CMP/MEP populations. For analyses of SLAM-LSK and CLP, the cells were stained with PE-Sca1 (Ly-6A/E), Pacific Blue-CD117 (c-Kit), APC-CD150 (SLAM), APC-eFlour780-CD48, and PerCP-Cy5.5-CD127 (IL-7Ra). Expression of CXCR-4 and integrins in the hematopoietic progenitor cells was evaluated by co-staining lineage negative cells or c-Kit+ HSPCs with PE-Sca1(Ly-6A/E), PE-Cy7-CD117(c-Kit), and APC-CD184 (CXCR4)/APC-CD11a/CD11b/CD18. The information for antibodies is provided in Supplementary Table 2.

All staining was carried out at room temperature for 15–20 min. The samples were then washed with FACS buffer and kept on ice until FACS analysis. The absolute cell number was acquired by incorporating CountBright Absolute Counting Beads for flow cytometry (Catalog# C36950, Thermo Fisher Scientific). Gating of murine bone marrow HSPC subpopulations was performed as described previously[26,27], and was further illustrated in Supplementary Fig. 8. Flow cytometric analysis was performed with a BD LSRFortessa cell analyzer, BD FACSCanto II, or BD LSRFortessa X-20 flow analyzer, and further analyzed with FlowJo software 10.3.0 (TreeStar Inc.).

**In vitro HSPC expansion.** c-Kit+ HSPCs from the bone marrow were purified using a mouse CD117 (c-Kit)-positive selection kit (STEMCELL Tech.) according to the manufacturer's instructions. The purified c-Kit+ cells were further cultured in StemSpan serum-free expansion medium (SFEM) (Catalog #09650, STEMCELL Tech.) supplemented with 10 ng/ml IL-3 (Catalog #300-324P, GEMINI Bio-Products.),10 ng/ml IL-6 (Catalog #300-327P, GEMINI Bio-Products.), 50 ng/ml SCF (Catalog # 300-348P, GEMINI Bio-Products.), and human LDL (1:600–1:1000) (Catalog #02968, STEMCELL Tech.) for 24 h before transduction or further culture as indicated in the figure legends.

**Generation of retroviral particles and transduction of HSPCs.** To generate retroviral particles, HEK293T cells were seeded in 10 cm culture dishes at $6 \times 10^6$ for 16–18 h in high-glucose DMEM (Catalog #10-013-CM, CORNIING) containing 10% FBS (Catalog #900-108, GEMINI Bio-Products.), followed by co-transfection of 9 μg MICD4-based retroviral constructs and 4.5 μg of the packaging construct pCL-Eco with TransIT-LT1 transfection reagent (Mirus) according to the manufacturer's protocol. Viral supernatants were collected 48 h after transfection and exchange with fresh medium. After an additional 24 h, all virus-containing supernatants were pooled. The debris was removed by brief centrifugation. The virus supernatants were concentrated using a 100 kDa cutoff Amicon Ultra-15 Centrifugal Filter Unit (Catalog #UFC910024, EMD Millipore) through centrifugation at 4 °C following the manufacturer's instruction. Retroviral infection of c-Kit+ HSPCs was performed by suspending the cells in freshly concentrated viral supernatants in the presence of 8 μg/ml polybrene (hexadimethrine bromide, Catalog #H9268, Sigma) and centrifuged at $900 \times g$ for 90 min at 37 °C. After spin-infection, the viral supernatants were gently removed and the cells were incubated with fresh SFEM containing cytokines and further cultured for 36–48 h in vitro for further experiments as indicated for subsequent experiments.

**Quantitative real-time RT-PCR and RT-PCR.** RNA isolation, complementary DNA synthesis and quantitative real-time PCR were performed as previously described[25–27]. Briefly, total RNAs from c-Kit+ HSPCs were extracted with TRIzol (Catalog #15596018, Invitrogen) according to the manufacturer's protocol. cDNA was reverse transcribed with 1 μg total RNAs by qScript cDNA supermix (Catalog #84034, Quanta Biosciences) as described by the manufacturer. Synthesized cDNA samples were amplified in a StepOnePlus Real-Time PCR System by using

PerfeCTa SYBR Green QPCR FastMix with ROX (Catalog #95073-012, Quanta BioSciences) in triplicate. The default cycling conditions were 95 °C for 5 min and 40 cycles of 95 °C for 15 s, 60 °C for 60 s. Melting curve analyses were performed at the end of the reaction to confirm amplification of a single PCR product. Cycle threshold (Ct) values were calculated with StepOnePlus software (v2.3). The amplification efficiency of each pair of primer was determined by relative standard curve experiments. Relative target gene expression levels were further assayed by quantitative real-time RT-PCR with the comparative $C_T$ ($\Delta\Delta C_T$) method and were normalized to 18S rRNA (eukaryotic 18S ribosomal RNA) levels in each sample. Results are expressed as mean ± SEM.

The RT-PCR procedure for detecting SRF isoforms was performed as previously described[69]. Briefly, total cellular RNAs from FACS-sorted LK and LSK cells were reverse transcribed by qScript cDNA SuperMix following the manufacturer's protocol. PCR was performed in a separate tube with 5 μl of cDNA as a template. The 50 μl PCR reaction mix contained standard PCR buffer with 1.85 mM MgCl$_2$, 0.2 mM deoxynucleoside triphosphates, sense and antisense primers (each 0.4 μM), and platinum $Taq$ polymerase (Life Technologies). The cycling conditions included an initial 3-min 95 °C denaturation, followed by 50 cycles of denaturation for 15 s at 95 °C, annealing at 50 °C for 55 s, and a 3 min extension at 72 °C. A 10 μl aliquot of reaction mix was fractionated on a 2% agarose gel, stained with ethidium bromide, and photographed. Negative controls lacked cDNA templates in the reaction mix. The primer sequences are listed in Supplementary Table 1.

**Adhesion assay.** c-Kit+ HSPCs purified from control or mDia2-deficient mice were incubated with HSPC expansion medium (SFEM) containing cytokines for 3 days. The suspended cells were applied for adhesion assays with collagen (H-22)-coated or fibronectin (GG-22)-coated coverslip (Neuvitro) for 1 h at 37 °C, or on poly-D-lysine (CORNING, REF344085)-coated coverslips for 15 min at 37 °C. After washing with PBS twice, the cells were fixed and stained with DAPI. For ICAM-1-binding assays, the coverslips were coated with recombinant mouse ICAM-1 (Gln28-Asn485) (Catalog# 796-IC, R&D) following the manufacturer's instructions, and incubated with cells for 2 h at 37 °C. After washing with PBS, the cells were fixed and stained with DAPI. 5–10 random fields were selected in each coverslip with duplication for cell counting under an inverted fluorescence microscope at a magnification of 200×. The adhesion ability of indicated cells was normalized with respect to the control.

**In vitro trans-endothelial migration of HSPCs.** Murine EC (Catalog #C57-6221) and the complete culture medium (Cat# M1168) were purchased from Cell Biologics (Chicago, USA). EC ($6 \times 10^4$) were seeded on the upper chamber of a Transwell (5 μm aperture, coated with 0.1% gelatin) for 72 h. Endothelial cell monolayer formation was visualized and examined with crystal violet staining. The endothelial cell complete medium was then removed and the cells were washed twice with PBS prior to loading HSPCs enriched in Lin− cells (~$6 \times 10^4$) in serum-free IMDM. After incubating for 4 h at 37 °C, cells that had migrated to the lower chamber with or without 200 ng/ml SDF1α and 1% input cells were collected and used in a colony formation assay as described above except that erythropoietin was not added. The number of HSPCs that had migrated was calculated as the percentage of colony-forming cells (CFC) in the lower chamber normalized to CFC generated from input cells.

**Bone marrow transplantation.** Non-cBMT and cBMT were performed as described previously[25]. Briefly, the recipient mice (8–9 weeks old) were pre-conditioned by the treatment of antibiotic-containing water (1.1 mg/ml neomycin and 2000 U/ml polymyxin B, Sigma) for one week. The mice were then lethally irradiated (1000 rad) followed by retro-orbital injection of BMMC ($2 \times 10^6$ for

BMT, $2 \times 10^6$ plus an equal number of CD45.1+ BMMCs for cBMT) or c-Kit+ cells ($1 \times 10^6$ for BMT, $1 \times 10^6$ mixed with an equal number of CD45.1+ competitors for cBMT). For serial transplantation, $2 \times 10^6$ donor BMMCs from primary transplant recipient mice were transplanted into lethally irradiated mice unless otherwise indicated. All the recipient mice continued to receive water containing antibiotic for 3 weeks, after which they were given regular water. Flow cytometry of peripheral blood was performed to assess engraftment. Animal survival was monitored throughout the experimental period.

For the intrafemoral injection, mice were anesthetized by isoflurane inhalation. The left or right knee joint was sterilized through three rounds of 70% alcohol wrap. Cells isolated from the bone marrow as described above were injected into the joint by puncture with a 22-gauge needle.

**Complete blood cell counts.** Peripheral blood samples were obtained by retro-orbital (RO) bleeding and stored in Greiner MiniCollect EDTA tubes containing K3EDTA (Catalog #450475). Complete blood cell counts were assayed using a Hemavet 950 instrument (Drew Scientific).

**Cell quiescence analysis.** Stem cell quiescence was profiled by pyronin Y or Ki67 staining. Briefly, HSPC-enriched lineage-negative cells from the bone marrow were labeled with PerCP-Cy5.5-Sca1 and APC-CD117 (c-Kit), washed and re-suspended in Iscove's modified Dulbecco's medium (IMDM; Catalog #12440-046, Thermo Fisher Scientific) containing 10 µg/ml Hoechst 33342. The cells were incubated for 45 min at 37 °C. Pyronin Y solution (100 µg/ml in PBS; Catalog #213519, Sigma) was then added directly to the cell suspension to a final concentration of 0.5 µg/ml and incubated for another 15 min. The cells were then transferred onto ice before assessment by flow cytometry. Nuclear staining of Ki67 was performed by using an FITC-anti mouse/Rat Ki-67 monoclonal antibody (SolA15) (Catalog #11-5698-82) and a Foxp3/transcription factor staining buffer set (Catalog #00-5523-00, eBioscience). In brief, after initial staining with PE-Sac1 and APC-CD117 (c-Kit), the cells were fixed and permeabilized with Foxp3 Fixation/Permeabilization solution for 30–60 min at room temperature. The cells were then washed and blocked with anti-rat serum overnight followed by staining with FITC-anti mouse/rat Ki-67 monoclonal antibody (SolA15, 1:1000; Catalog #11-5698-82) for 1 h in the dark at room temperature. Prior to analysis, cells were incubated with Hoechst 33342 for 30 min at room temperature.

**Homing assay.** BMMCs ($2 \times 10^6$) or bone marrow lineage negative cells (CD45.2+, $2 \times 10^7$) from control or mDia2-deficient mice were mixed with an equal number of CD45.1 competitive BMMCs or lineage-negative cells. The cells were then injected retro-orbitally into lethally irradiated WT recipient mice that were sacrificed at 24 or 48 h later for flow cytometry of BMMCs. Lineage-negative cells were harvested from the tibia and femur 12-h post injection for FACS analysis of c-Kit+ HSPCs. The population of CD45.1+ versus CD45.2+ cells was determined.

**Immunofluorescent staining.** Sorted c-Kit+ HSPCs were suspended in serum-free IMDM or IMDM containing 10% FBS and plated onto poly-D-lysine-coated cov-erslips, followed by incubation at 37 °C in a humidified incubator for 10 min. The attached cells were washed with ice-cold PBS, fixed in 4% paraformaldehyde for 15 min, and permeabilized with 0.1% Triton X-100 in PBS for 10 min at room temperature. After rinsing three times in PBS, the cells were blocked with 3% BSA in PBS with 0.05% Triton X-100 for 1 h at room temperature. The cells were then stained with Alexa Fluor 594 Phalloidin (165 nM, Catalog #A12381, Thermo Fisher Scientific) and Alexa Fluor 647 anti-Tubulin-α antibodies (1:300, Catalog #627908, BioLegend) for 1 h at room temperature, followed by three 5-min washes with PBS. After the final wash, the cells were incubated with 1 µg/ml DAPI to stain the nuclei followed by PBS washing. For MAL/Lamin B1 staining, after an overnight incubation with primary antibody (Anti-MAL, 1:200, sc-390324, Santa Cruz Bio-technology; anti-Lamin B1, 1:200, ab16048, Abcam), the cells were further stained with Alex Fluor 594 goat anti-mouse IgG (1:400) and Alex Fluor 647 donkey anti-rabbit IgG (1:400) for 1 h in the dark at room temperature followed by three washes with PBS. The coverslips were then mounted on a glass slide with Slowfade Antifade reagent (Invitrogen). The images were acquired with a Nikon A1R laser-scanning confocal microscope and were further analyzed by NIS Elements soft-ware. The information for antibodies is provided in Supplementary Table 2.

**G-CSF injection.** Recombinant human granulocyte colony stimulating factor (G-CSF) (Catalog #300-123P, Gemini Bio-Products) was diluted in endotoxin-free PBS containing 0.1% bovine serum albumin (Catalog #09300, StemCell Tech.). mDia2 conditional knockout and control mice were intraperitoneally injected with 5 µg G-CSF for 5 consecutive days and sacrificed 12 h after the last injection. Lineage-negative cells were purified from the bone marrow, spleen and peripheral blood, and subjected to antibody staining with PE-Sca1 and PE-Cy7 CD117(c-Kit) for FACS analysis. Colony formation assays were performed using nucleated cells from the peripheral blood. Colony numbers were scored using an inverted microscope and calculated on the basis of colony number per milliliter of peripheral blood. Cells from all colonies per well were pipetted with 1× PBS, centrifuged, and re-suspended for FACS assay to determine the cell numbers. The data are presented as the number of cells derived from colonies per microliter of peripheral blood.

**Colony-forming unit assay.** Cells were cultured in 1 ml of methylcellulose med-ium (Methocult M3234, STEMCELL Tech.) containing 50 ng/ml SCF, 10 ng/ml IL-3, 10 ng/ml IL-6, 3 Unit/ml erythropoietin (EPO, Catalog #GH002, HumanCells Bio.), and 1× penicillin and streptomycin in six-well plates. After incubation at 37 °C with 5% $CO_2$ and high humidity for 7 days, the colony-forming units (CFUs) were scored according to manufacturer's instructions with an Olympus CKX31 inverted microscope.

**Apoptosis analysis.** Apoptosis was assayed by staining freshly harvested cells with lineage, stem, and progenitor markers as described above, followed by Cy5-Annexin V (Catalog #559934, BD Pharmingen) or Pacific blue-Annexin V (Catalog #640918, BioLegend) staining (1:200 diluted in Annexin V-binding buffer) at room temperature for 15–20 min. DAPI or PI were added prior to initiating the flow cytometric assay.

**In vivo imaging.** For in vivo-imaging experiments, lineage-negative cells were purified form control and mDia2-deficient mice and labeled with the cell pro-liferation dye eFluor 670 (Catalog #65-0840-85, eBioscience) according to the manufacturer's instructions. After washing with completed medium containing 10% serum, labeled cells were mixed with an equal number of CD45.1+ lineage-negative cells and injected into lethally irradiated CD45.1+ mice via retro-orbital injection. At 18 h after transplantation, the mice were sacrificed, stabilized on a custom-made pedestal and the femoral bone was carefully shaved to expose the bone marrow. One hour before sacrifice and the initiation of the imaging protocols, mice were retro-orbitally injected with PE-CD31 Ab (3 µg/ml, Catalog #12-0311-82, Thermo Fisher Scientific) in 100 µl Hank's balanced salt solution (HBSS) to outline endothelial sinusoids and (or) arterioles in the bone marrow. Imaging was performed using an UltraVIEW VoX imaging system built on an Olympus BX-51WI Fixed Stage illuminator and equipped with a Yokogawa CSU-X1-A1 spin-ning disk, a Hamamatsu EMCCD C9100-50 camera, and a Modular Laser System with solid state diode lasers with DPPS modules for 488, 561, and 640 nm and the appropriate filters (all assembled by Perkin Elmer, Naperville, IL). For each field, Z-stacks of 40 images with step intervals of 2.0 µm (80 µm depth) were acquired using a Piezoelectric MIPOS100 System (Piezoystem Jena, Germany) controlled objective positioning. Images were collected using a ×20 water-immersion objective (1.00 numerical aperture). Volocity® software (Perkin Elmer, v6.1) was used to drive the microscopy and acquisition of images, which were then analyzed using ImageJ (1.51m9).

**Luciferase constructs with *ItgaM* and *Itgb2* intronic regions.** Genomic DNA from wide type mouse tails was extracted by precipitation using two volumes of ethanol after Proteinase K (EO0492, Thermo Fisher Scientific) digestion. Genomic DNA (~50 ng) was used for PCR amplification with Platinum Hot Start PCR Master Mix (Catalog #13000012, Thermo Fisher Scientific) or CloneAmp HiFi PCR Premix (Catalog #638500, Takara). Thermal cycling conditions were dena-turation at 95 °C for 10 min, followed by 32 PCR cycles of 95 °C for 15 s, 65–67 °C for 45 s and 72 °C for 1.5 min with a final 10 min elongation step at 72 °C. DNA fragments of the *ItgaM* intron 2 and *Itgb2* intron 6 regions (each ~0.7 kb) were then sub-cloned to the pGL4.24 (luc2P/minP) vector (E842A, Promega) between the NheI and HindIII sites using the In-Fusion HD Cloning System (Catalog #638909) according to the manufacture's protocol. All the constructs were vali-dated by DNA sequencing.

**Site-directed mutagenesis or deletions.** To generate site directed and deletion mutants, PCR in a 20 µl reaction volume containing 10 ng of plasmid DNA tem-plate, 20 ng primer pairs and 0.5 U AccuPrime *Pfx* DNA polymerase in 1× reaction mix was performed. PCR amplification was initiated with 2 min at 95 °C to denature the template DNA, followed by 18 amplification cycles for 30 s each cycle, 55 °C for 1 min and 68 °C for 5–8 min depending on the template construct length (1 kb/min for *Pfx* DNA polymerase). The PCR products were treated with 0.5–1 µl of FastDigest *DpnI* (FD1703, Thermo Fisher Scientific) at 37 °C for 45 min and 8 µl of each PCR product was further analyzed by agarose gel electrophoresis. 5 µl PCR products generated above were transformed into *E. coli* competent cells. All site directed and deletion mutants were verified by DNA sequencing.

**Dual luciferase activity assays.** HEK293T cells ($1.25 \times 10^5$ cells per well) were incubated in 48-well plates for 18–24 h until the cell density reached 70–80% confluence prior to transfection. Cells in each well were then co-transfected with pGL4.24 (luc2P/minP) vector alone or with different *Itgb2* and *ItgαM* intronic fragments (100 ng per well), SRF expressing constructs (MICD4–mSRF, MICD4–mSRFΔ5, or MICD4–HA human SRF VP16, 400 ng per well in Fig. 4c, f, g; increasing amount of 100, 200, and 400 ng per well in Fig. 4d) or empty vector (MICD4), as well as the Renilla luciferase expression vector pRL-TK (1 ng per well in Fig. 4c, f, g and 0.5 ng/well in Fig. 4d) using TransIT-LT1 reagent (Mirus; 1:3 DNA: Reagent, 1.5 µl). Total amounts of DNA in each well were normalized relative to the empty vector MICD4. After 24 h, the cells were harvested, and luciferase activity was measured using the Dual Luciferase reporter assay system (E1910, Promega) according to the manufacturer's instructions. Firefly luciferase activities were normalized to *Renilla* luciferase activities (Firefly Luc/Renilla Luc)

and calculated as the fold-change from empty vector. All the luciferase experiments were performed in duplicate, and the data are presented as mean ± SEM from 3 to 4 independent measurements using separate luciferase constructs from two or three different clones.

**Rapid measurement of sgRNA activity by luciferase reporter.** To establish a rapid and quantitative method to reliably evaluate genome-editing activities of various sgRNAs in mammalian cells, we built a luciferase activity-based reporter assay system. Although editing-based disruption of constitutively expressed fluorescence markers, such as GFP and tRFP657 have been widely applied to test sgRNA activity[70], this approach involves a long detection period of 8–14 days. As such, decreases in fluorescence signal intensity do not occur immediately, even with the genetic disruption of the fluorescent gene locus because of the half-life of the remaining intact transcripts and turnover due to protein degradation[70–72]. To overcome this delayed effect, we took advantage of the rapid response of the pGL4.24 luciferase construct (Promega), in which the PEST sequence, a protein degradation signal from the C-terminal region of mouse ornithine decarboxylase, is incorporated into the synthetic firefly *luc2* gene (*luc2P*). Specifically, sgRNAs were designed online using the CCTop-CRISPR/Cas9 target online predictor or Design sgRNAs for CRISPRko from the Broad Institute. The sgRNAs were synthesized and ligated into the sgRNA expression vector BPK1520 (Addgene #65777) or pLKO5. sgRNA.EFS.tRFP657 (Addgene #57824). Potential sgRNA-targeting sites were selected and synthesized as gBlock DNA fragments (IDT), which were further PCR amplified and in-frame fused with the *luc2P* gene between the *NcoI* and *ApaI* sites by In-fusion cloning (Fig. 6a). To establish a *Streptococcus pyogenes* Cas9 over-expression stable cell line, HEK293T cells were transduced with pLenti-Cas9-BSD (Addgene #52962) lentivirus and subsequently selected with 10 μg/ml blasticidin (Thermo Fisher Scientific, R21001) to obtain single clones. HEK293T-4# mono-clonal cells showed strong FLAG-Cas9-NLS expression and consistent efficiency and were used as a rapid mammalian genome-editing reporter system for all subsequent assays. The newly designed target gene genome mimic luciferase construct (100 ng per well) was transiently transfected into HEK293T-4# cells together with various sgRNA vectors (300–400 ng per well) in the presence of pRL-TK (1 ng per well). Luciferase activity was measured 24 h post-transfection. When screening sgRNAs that target SRE sites in the mouse *ItgaM* intronic region, we observed a significant suppression of the luciferase activity by the incorporated intronic region but not the exonic sequences. This inhibitory effect was depressed by adding DNA sequences encoding 3× (GSSSS) amino acids between the intronic target sites and the *luc2P* gene[73] (Supplementary Fig. 6c, d).

**ChIP assay.** Purified c-kit+ bone marrow HSPCs were incubated with or without 10% FBS in IMDM for 30 min and then collected in formaldehyde. ChIP assays were performed using the Pierce Magnetic ChIP Kit (Catalog #26157, Thermo Fisher Scientific) according to the manufacturer's protocol. Briefly, chromatin was incubated with anti-SRF antibody (2C5, Catalog #61386, Active Motif), anti-MAL antibody (sc-390324, Santa Cruz), or anti-GFP (B2, Catalog #sc-9996, Santa Cruz) at 4 °C overnight. DNA was purified from the washed chromatin using a DNA extraction kit followed by reversal of crosslinking and proteinase K digestion. Immunoprecipitated DNA was analyzed by qRT-PCR with specific primer pairs spanning the putative SRF-binding sites present in the *ItgaM* and *Itgb2* genome regions or *Acta2* gene locus. Primers for the LINE1 gene were used as an internal control. The primer sequences are listed in Supplementary Table 1. The relative binding enrichments were calculated and normalized relative to anti-GFP immunoprecipitation of control cells without serum stimulation.

**Genome editing in the hematopoietic cells in vivo.** To edit mouse genomic DNA in vivo, lentiviral particles were prepared by co-transfection of 9 μg pLKO5.sgRNA. EFS.tRFP657-sgRNA expression constructs together with 4.5 μg of each of the packaging plasmids (VSVG/pMDL/REV) into 293T cells in 10-cm plates by TransIT-LT1. Viral supernatant collection, virus concentration and subsequent spin infection were then performed as described above except that c-Kit+ HSPCs purified from CAG-Cas9 transgenic mice were used. At 36–48 h after recovery, the cells were collected and used for BMT into lethally irradiated CD45.1 congenic mice. The percentages of tRFP657/GFP double-positive cells in the peripheral blood were evaluated by flow cytometry to determine the engraftment efficiency 1 or 2 months after transplantation. Loss of integrin expression in sgRNA-transplanted mice was confirmed either in circulating granulocytes (Fig. 6d) by staining PBMCs with PE-Cy7-Gr1, PE-CD11b, and BV421-CD18, or in HSPCs by co-staining lineage-negative cells with PE-Cy7-CD117, PE-Sca1, V500-CD11b, and Pacific blue-CD18 (Fig. 6e).

To determinate non-homologous end joining (NHEJ)-mediated indel mutations and repair of the *ItgaM* intron 2 region containing SRE during Cas9-induced genome breaks, genomic DNA was isolated from the peripheral blood and the target regions were PCR-amplified. The PCR products were directly cloned into the pCR2.1 vector (Catalog #K204001, Thermo Fisher Scientific) and subjected to Sanger sequencing to estimate NHEJ frequencies (Supplementary Fig. 6d).

**Statistics.** Results are expressed as mean ± SEM unless otherwise indicated. Statistical comparisons between two groups were performed with a two-tailed

unpaired Student's *t* test using GraphPad Prism version 8.0 software. Survival curves were compiled using Kaplan–Meier algorithms in Prism software, and the significance was assessed using Mantel–Cox log-rank test. $p < 0.05$ was considered statistically significant.

**Reporting summary.** Further information on research design is available in the Nature Research Reporting Summary linked to this article.

## Data availability

The data that support the findings of this study have been uploaded online as source data file. The Gene Expression Commons database was open-accessed upon registration (https://gexc.riken.jp). The DNA sequences from ChIP-seq peak were recalled by UCSC genome browser (MM8), and the location of the binding peaks were further illustrated within the genomic locus from open-accessed Ensemble database in Fig. 4a. The gene expression and associated epigenetic information in HSC as shown in Supplementary Fig. 5d, e were extracted from the public HSC Aging Hub (http://dldcc-web.brc.bcm.edu/lilab/AgingHSCEpigenome/browser.html). Source data underlying Figs. 1b, c, e, 2c, d, f, g, i, k, 3c–e, h, j–l, 4c–j, 5a, c, e–g, 6b–f, h, i and Supplementary Figs. 1a, d, f, 2a–d, g, i, 3a–c, f, 4b–d, 5b, c, 6a, b, d, and 7a–f are available as a Source Data file. Other data that support the findings of this study are available from the corresponding author upon reasonable request. Source data are provided with this paper.

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

## Acknowledgements

We thank Baobing Zhao (Shangdong University, China) for technical assistance, Matthew DeBerge, Edward Thorp, and William A. Muller for sharing the fluorescence microscope and other resources, and Naren Ramanan for providing the SRF-VP16

construct. We are grateful to the Northwestern University Center for Advanced Microscopy core for the help with the imaging experiments and Zheng Jenny Zhang for the help with intrafemoral injection. This work was supported by National Institute of Diabetes and Digestive and Kidney Disease (NIDDK) grant R01-DK102718 (P.J.), R01-DK124220 (P.J.), National Heart, Lung, and Blood Institute grant R01-HL148012 (P.J.) and Department of Defense grant CA140119 and CA180214 (P.J.). X.H. is a recipient of an American Heart Association postdoctoral research award. P.J. is a recipient of a Leukemia and Lymphoma Society Scholar Award and a Harrington Discovery Institute Scholar Award.

## Author contributions

Conceptualization, Y.M. and P.J.; Methodology, Y.M., X.H., R.S., and P.J.; Investigation, Y.M., X.H., Y.L., J.Y., and R.S.; Writing—original draft, Y.M. and P.J.; Writing—review & editing, Y.M. and P.J.; Funding acquisition, P.J.; Supervision, P.J.

## Competing interests

The authors declare no competing interests.
