## [Peer Review File · Nature Communications]

Reviewers' comments:

Reviewer #1 (Remarks to the Author):

Summary:

While actin cytoskeletal remodeling is important in hematopoietic cell motility and migration, the precise mechanisms by which hematopoietic stem and progenitor cells (HSPCs) establish residency following homing to the bone marrow (BM) niche remain unclear. Two major protein families regulate actin-nucleation and cellular motility in mammalian cells, (1) formin-related proteins and (2) WASP protein complexes. mDia1, mDia2, and mDia3 belong to a highly conserved family of diaphanous-related formins proteins that modulate a number of cell processes in addition to actin polymerization, including the modulation of microtubule networks, mRNA trafficking, serum response factor (SRF)-mediated gene expression, and cytokinesis. In this manuscript, Mei et. al. explore mechanisms of mDia2 (Diaph3) and, to a lesser extent, mDia1 (Diaph1)-mediated HSPC transmigration and engraftment into the HSPC-supportive bone marrow microenvironment using in vivo knockout strategies paired with genetic complementation to rescue HSPC deficiencies in a bone marrow transplantation (BMT) model system. In competitively transplanted recipients, loss of mDia2 (using a hematopoietic-specific Vav1-cre and Mx1-cre) results in distinct hematopoietic failure in non-competitive and competitive BMTs. Interestingly, impaired HSPC function is not due to a homing defect, but rather a failure of HSPCs to properly transmigrate from the peripheral blood to the BM parenchyma, and vice versa. This was confirmed using a combination of BM imaging following BMT, in vivo hematopoietic mobilization assays, and ex vivo migration assays of mDia2 knockout HSPCs. mDia1 knockout animals display a significant, but more mild engraftment phenotype (when compared to mDia2 knockouts). HSPC transmigration was not examined in mDia1 knockouts. Downstream of mDia2 activity, MAL-SRF signaling displayed functional deficiencies including a decrease in some canonical SRF targets and non-canonical beta2 integrins CD11a, CD11b, and CD18 targets. The authors confirmed enhancer binding activity in both CD11b and CD18 integrins using mutational enhancer analysis in a luciferase assay. Genetic complementation (overexpression) of SRF in mDia2 knockout HSPCs restored the observed engraftment phenotype. Moreover, the authors confirmed the function of SRF-dependent integrins by deleting CD11b and CD18 using CRISPR-Cas9 phenocopying the mDia2 knockout BMT mice. Genetic complementation (overexpression) of CD11b in mDia2 knockout HSPCs also partially restored the observed engraftment phenotype. The approach to examining the mechanism of mDia2 loss of function in HSPC engraftment in this manuscript is exhaustive and provides a significant springboard for future studies. The findings also provide significant insights into HSPCs migration and lodgement phenotypes, and should be of interest to the general readership of Nature Communications. The inclusion of data from mDia1 knockout mice is some cause for concern: (1) The phenotype in mDia1 mice are not examined in detail (compared to mDia2 knockout mice), (2) Analysis of mDia1/mDia2 double knockouts is incomplete, and (3) the mDia1 knockout model is a global knockout that may have additional niche-mediated effects on HSPCs. Moreover, the discussion points out a number of obvious questions that can be explained by mDia1 overlapping function. The inclusion of mDia1 in this story demands experimental answers to these questions (e.g. expansion of HSPCs phenotype in SRF knockouts, page 19 and partial SRF target gene dysregulation in mDia2 knockouts, page 20, and proper HSPC experimental phenotyping/MAL-SRF signaling investigation). As currently presented by the authors, the manuscript is incomplete. If the authors choose to leave mDia1 knockout model in the manuscript, they should address the questions raised in Major Points #5. Alternatively, this manuscript can focus on the mDia2 story and address mDia1 more completely in another manuscript.

Major Points:

1. Can the authors explain the rationale for examining non-competitive BMT (Fig 1) at eight weeks

post-transplantation (as opposed to standard 16-24 week time points in which homeostatic hematopoiesis reestablished)? Considering the dramatic engraftment phenotype in Vav1-cre; mDia2fl/fl competitive BMT (Fig 2), does engraftment of HSPCs in Mx1-cre; mDia2fl/fl in non-competitive BMT represent non-recombined HSPCs?

2. The initial description and experimental usage of mouse models in this manuscript is confusing. Model generation: The methods and text do not adequately describe the generation of these models. Please provide source (catalog/vendor) information for the Mx1-cre and Vav1-cre. Is the Vav1-cre constitutively active during development or an inducible model? Experimental usage: What is the reasoning for utilizing both Mx1-cre and Vav1-cre models interchangeably when making hematopoietic comparisons (e.g. steady state Sup 1C, E vs BMT stress Fig 1A). What is the significance of comparing results from the two models (i.e. what new information does it tell the reader)? Why was Mx1-cre and Vav1-cre chosen for this study? In some cases, data is left out entirely for comparisons between models (e.g. page 6, Mx1-cre steady state analysis).

3. The observation that HSPCs fail to transmigrate into the in vivo bone marrow niche (Fig 2G-H) is critical piece of information for this manuscript. In addition to the in vitro migration assay, it directly implicates a lodgement phenotype. However, the images (Fig 2G) do not give the reader any magnitude of this observation. It is important to have a much more representative series of images to support the quantification (Fig 2H). This data can be amended as a supplementary Figure. Scale bars also needs to be added to all microscopic images. It would also be beneficial to test this hypothesis; does intrafemoral injection of mDia2 knockout HSPCs circumvent the observed engraftment phenotype?

4. No cell cycle differences were observed in steady state mDia2 knockout. Are there any cell cycle differences noted following hematopoietic recovery in non-competitive mDia2 BMT mice (Fig 1A-C)?

5. The interplay between mDia1 and mDia2 in this manuscript are very interesting, but raise a series of questions concerning their mutual (compensatory) and mutually exclusive functions that should be addressed experimentally. (1) mDia2 functions through the MAL-SRF signaling axis, but only alters a subset of canonical/integrin SRF targets. Are the remaining targets altered in mDia1 knockout animals? (2) Is HSPC frequency and absolute numbers altered in under steady state and BMT conditions? (3) What is the status of canonical/integrin SRF targets in the mDia1/mDia2 double knockout HSPCs? (4) As noted by the authors, SRF knockout mice have a significant expansion of phenotypic HSPCs at steady state that was not observed in mDia2 knockouts. Do mDia1/mDia2 double knockouts display HSPC expansion under homeostatic conditions? (5) Does SRF rescue mDia1 single knockout or mDia1/mDia2 double knockout HSPCs? (6) Do mDia1 knockout mice display an transmigration defect similar to mDia2 knockouts (imaging, mobilization, Transwell assays)? (7) Is the transmigration phenotype more pronounced in mDia1/mDia2 double knockout HSPCs?

Minor Points:

1. The intermittent inclusion of heterozygous deletion of mDia2 (e.g. Fig 2C) is distracting and does not add any value to the conclusions of the paper.

2. Figures use a combination of listed p-values and asterisks in bar graphs to indicate significance. Please correct for consistency.

3. Figure legends should have indications of which statistics were used to generate p-values.

4. Include significance for engraftment in Fig 2C.

5. Include significance for engraftment in Fig 7A.
6. An ANOVA analysis (repeated measures) may provide statistical power to Fig 7C.
7. The Methods sections lacks experimental detail in many sections (e.g. Clear explanations the mice/models used, how were the animals preconditioned for BMT, etc.). Please ensure transparency and clarity for the readers.

Reviewer #2 (Remarks to the Author):

Homing and establishment of HS/PCs in the bone marrow is critical for establishment of hematopoiesis, but the molecular mechanisms involved are poorly understood.

In this study, Mei et al investigate the role of mDia1 and mDia2 formins in bone marrow engraftment of hematopoietic stem and progenitor cells (HSPCs). First they show that mDia2 KO cells show defective reestablishment of LSK and LT/ST-HSC populations, with LSK KO cells showing loss of quiescence. Similar but weaker phenotypes were seen with mDia1 KO., which appears to synergise with mDia2. They show that mDia2KO did not affect homing of the HSPC to the bone marrow vasculature but led to impaired transendothelial migration and niche engraftment, and defective mobilization by GCSF. This was associated with inactivation of the MAL transcriptional coactivator, which became refractory to serum stimulation, and reduced expression of MAL-SRF target genes. The authors provide evidence that Itgb2 and ItgaM are direct targets for SRF. They go on to show that retroviral transduction of SRF restores classic SRF target gene expression as well as integrin expression, adhesion, and engraftment.

Next they use CRISP/Cas of ItgaM or Itgb2 to show that these integrins are required for recruitment, their combined deletion being synergistic. Ectopic expression of ItgaM can partially rescue the mDia2KO phenotype.

EVALUATION

The paper nicely shows that MAL-SRF signalling is downstream of mDia2 in HSCP engraftment, and that integrin expression is an important aspect of this with ItgaM and Itgb2 playing important roles. It also shows redundancy with mDia1. What it does not do is convincingly show that MAL-SRF signalling activates the integrins directly, as opposed to indirect effects arising through MAL-SRF's well known role in regulating expression of multiple cytoskeletal genes. The paper is marred by an unconvincing analysis of the integrin regulatory sequences and particularly poor presentation of this section.

MAJOR ISSUES

Ectopic expression of ItgaM can partially rescue the mDia2KO phenotype, but Itgb2 is not tested, alone or in combination with ItgaM. What happens with Itgb2 ± ItgaM? Moreover, they authors do not test ItgaL either. These experiments should be done and the results shown.

A critical point of the authors' model is that mDia2KO cells should be defective in recruitment of MAL to SRF binding sites (ANY SRF binding sites that recruitment MAL). This can be demonstrated by ChIP, but no data are provided. This is essential to validate the proposed integrin SRF binding sites, and for canonical target genes such as Acta2 or SRF itself.

The SRF binding analysis is not adequately presented and does not cite the appropriate literature.

OTHER POINTS (not in order of importance)

Order of presentation. Two points:

- The presentation of the Dia1 data occurs in Figure S3(p8), then in Figure 7 (p17), and as a result the Figure 7 data reads as an afterthought. Should present the mDia1 combination with mDia2 before dealing with mechanism.
- The Integrin deletion and rescue data – strong data that show that it is integrin expression is limiting upon mDiaKO – could be presented before the data showing that they are SRF targets.

Absolute cell counts should be displayed in Figure 1D

The transwell assay in Figure 2F could be performed on a monolayer of endothelial cells as well, which would better mimick the in vivo context and confirm whether the transendothelial migration of KO cells is defective in vitro

Figure 3E/F. The MAL and SRF and gene expression data is not convincing.

3E -The localisation figure needs quantitation, it is only shown as images of representative cells, but it should also be quantified over 50-100 cells in Figure 3E, with markers for the cytoplasm.

3F - some these genes are MAL-controlled and others by the Elk1 transcription factor but it looks like there are Elk targets affected by mDia2KO (eg Egr1) and MAL targets that are not (eg Vcl, Tgln). What is going on? What happens if the serum-stimulated cells in 3E are analysed by RT-PCR?

3F – relative expression data do not agree with Costello et al. which estimated relative WT/KO as Itgb2: 1.14; Itgal: 1.01; Itgam: 0.75. Please comment.

Figure 3K – adhesion should also be performed on an ICAM substrate, which is one of the main physiological substrate of integrins alphaM/beta2

Figure 4 is not acceptable, and the authors appear to be unaware of the classical SRF literature.

- Figure 4A. The SRE sequence consensus is CC(A/T)6GG, which should be referenced properly. There is no good fit to this consensus in either of these two sequences. What do the arrows indicate?

- Figure 4C/D. These are not consistent. Why does SRF cause a 3fold effect on ITGAM2+ in C, but an 8fold effect in D?

- Figure 4F Assuming that the CCTTGAGAGG is a consensus, there is NO published literature supporting the notion that the C at position +6 downstream would have any effect on SRF binding – why was this mutant chosen? Where are two sites, and what is the evidence that they increase DNA binding activity?

- Figure 4G –Figure 4A bottom should be modified to match this. Where are the proposed SRE elements in this sequence, and why were the point mutations chosen? And what is a “G/C to T” mutation?

- Figure 4H. This experiment is not explained. What is the significance of the +10% FBS? SRF is meant to bind constitutively to its targets. Surely the authors should see a signal in the “-” lane too? A control with isotype-matched irrelevant antibody should be shown. What is the ChIP signal on a bona fide SRF target such as Acta2 or SRF itself? The authors argue that mDia2 controls MAL to regulate these integrins, so should also show a MAL ChIP. Finally a critical point in the authors’ model is that mDia cells should show a defect in MAL recruitment to SRF sites. This should be done.

- Figure 4I. The CRISPR/Cas experiment is not adequately explained in the text or figure legends. It appears that there are multiple edits in the population (Figure 4D), yet a single population is shown for CD11B analysis - explain. Moreover, the effect of the deletion should be evaluated by qRT-PCR on ITGAM2, not by CD11 MFI.

The competitive BMT experiment in Fig S3D made use of GFP labelled KO cells, tracking them 2 months post-transplantation; are cells infected with GFP expressing viral particles, or is GFP transgenically expressed? if cells were transduced, to control for the fact that the cells could have lost

the GFP during this time, the reverse labelling experiment should be performed, with unlabelled KO cells and GFP labelled WT cells; alternatively, a CD45.1/CD45.2 system could be used

Figure S3E The second competitive BMT here has not been initiated with a 1:1 mix, but with only 20% of KO cells, which seems to be the ratio at which they are found 6 months after the first cBMT; please clarify in the legend

Figures 2G&H It would have been good to have both WT & KO labelled with different dyes & injected into the same recipient. Limited information is presented - just two cells for the visual representation. The Figure needs to include details on numbers of cells counted per field, how many cells are there in 5-6 fields?

General Figure issues

- Statistical data (p-values and n) are missing in some panels, they should be added to all figure panels, and the statistical test used should be stated in the legends
- for the Kaplan-Meier survival curves, median survival for each condition (including control) should be clearly indicated in Figures 1F, 1G, 1H, 5J, 6G, 7B, 7H
- the y-axis should not be discontinued in graphs, it artificially amplifies small differences between conditions: please correct accordingly Figures 1C, 4J, S6A, S6E
- red and blue columns have been swapped in Figure 3H, please correct
- the legends of the x-axis (0 to 7 months post-cBMT) in Figure 7A should be shifted towards the left, consistent with pIpC injections one month after transplantation
- Figure S5, please specify in the legend what the blue shading indicates

Results p10: 'mDia2 regulates HSPC engraftment through SRF signaling'. This statement is too strong – these results shows that loss of mDia2 in HSPC results in defective F-actin, cell polarity, adhesion, MAL-SRF pathway and integrin beta2 expression, but not that this is cause of the engraftment defect. Please rephrase.

P10§3 should read "... reported to affect cell functions through INactivation of the MAL-SRF pathway in cell based assays"

reference 40 Is not about MAL. Replace.

Numerous typos – correct please.

Refs 27 and 66 are the same!

The comment in the discussion that transcription factors binding sites are located 5' of the TSS on promoter regions, and that an intronic binding site is unusual, is simply not true. Remove.

Reviewer #3 (Remarks to the Author):

In their manuscript, Mei et al. demonstrate that mDia1 and 2 -proteins involved in the regulation of actin- are involved in HSPCs homing, engraftment, and repopulation potential of lethally-irradiated mice. Whereas the transendothelial migration of leukocytes has been studied in detail before, this is not the case for HSPCs. Therefore, the manuscript substantially adds to the field. However, how mDia deletion impacts transplanted HSPCs is not so clear. Below are some points aimed at sharing clarity in this regard and improve the manuscript.

1) Endothelial cells lining BM sinusoids have been identified as critical for HSCs homing and engraftment [1]–[3] . However, it is well known that irradiation damages the niche, in particular sinusoidal endothelial cells [3]–[7]. Therefore it is unclear to what extent the findings reflect physiological migration rather than BM engraftment and reconstitution after myeloablation. Protocol in non-irradiated mice have been described [8]. Although the authors state that “the bone marrow vasculature remained intact”, Fig 2G-H shows disruption of BM vessels stained with CD31 which, together with increased vessel permeability, makes difficult to localize cells inside/outside of vessels. To address this, HSPC transmigration through vessels could be analyzed through intravital imaging in non-irradiated mice <15h after transplantation [8] for confirmation.

2) While it is clear that mDia1 and mDia2 have a role in the maintenance of transplanted HSCs and their repopulation potential, the mechanism seems unclear. mDia deletion could lead to impairment in transendothelial migration, or in the maintenance of HSPCs inside of their niche by altering their adhesion to specific ECM protein (like fibronectin as you describe) or specific stromal cells, or both. For instance, the effects of mDia on HSC might be triggered by defective adhesion (rather than/besides transmigration impairment), which could disrupt HSC quiescence leading to exhaustion. The different possibilities and the most important mechanism should be better demonstrated and explained. It would be interesting to assess if Itg B2 re-expression can restore the proliferation pattern in transplanted HSCs mDia2-deficient HSCs.

3) Fig 5: It would be better to show Itg surface expression by flow cytometry (rather than qpcr). Coculture assays of mDia2-deficient HSCs with endothelial cells (and not only fibronectin) with/without Itg B2 inhibition would help substantiate the contention that mDia2 regulates trans-endothelial migration through Itg B2.

4) The introduction can be improved. The authors state that the bone marrow repopulation is a two-step process, with homing occurring within 24h, followed by a long-term engraftment requiring lodgment to specific niches. However, the current bibliography indicates that immediately (1 hour) following a HSPC transplantation in lethally-irradiated mice the majority of donor cells locate in the central marrow region. This is followed by a rapid redistribution of the cells within 5h, resulting in a preferential seeding of LT-HSC-enriched cells closest to the bone surface, whereas more mature cells progressively engraft further away from bone [9], [10]. The bibliographic reference that is supposed to support the statement (ref 4 of the manuscript) is a review about the development of CXCR4 and VLA-4 inhibitors and how they may improve the utility and convenience of peripheral blood stem cell transplantation, which seems disconnected. Even if scarce, some studies regarding HSPC transmigration have been reported and should be presented in the introduction.

5) Fig 3H: Does the frequency of cells expressing these integrins change upon mDia2 deletion? It has been reported that VE-cadherin regulates hHSPCs trans endothelial migration [11]. Does VE-cadherin cell surface expression change in mDia2fl/fl Vav-Cre mice?

6) Fig 5A,F might be better placed in supplementary data for more streamlined presentation of the main data.

7) Describing more clearly the relationships between CD11a/CD11b/CD18 and Itgb2 in the manuscript would help the reader. Few typos should be corrected throughout the manuscript. This sentence needs to be rephrased: "how transendothelial migration of the HSPCs in vivo are rarely investigated".

[1] S. T. Avecilla et al., « Chemokine-mediated interaction of hematopoietic progenitors with the bone marrow vascular niche is required for thrombopoiesis », *Nat. Med.*, vol. 10, no 1, p. 64-71, 2004.

[2] J. M. Butler et al., « Endothelial cells are essential for the self-renewal and repopulation of Notch-dependent hematopoietic stem cells », *Cell Stem Cell*, vol. 6, no 3, p. 251-264, 2010.

[3] A. T. Hooper et al., « Engraftment and reconstitution of hematopoiesis is dependent on VEGFR2-mediated regeneration of sinusoidal endothelial cells », *Cell Stem Cell*, vol. 4, no 3, p. 263-274, 2009.

[4] B. O. Zhou, L. Ding, et S. J. Morrison, « Hematopoietic stem and progenitor cells regulate the regeneration of their niche by secreting Angiopoietin-1 », *eLife*, vol. 4, p. e05521, 2015.

[5] H.-G. Kopp et al., « Thrombospondins deployed by thrombopoietic cells determine angiogenic switch and extent of revascularization », *J. Clin. Invest.*, vol. 116, no 12, p. 3277-3291, 2006.

[6] X.-M. Li, Z. Hu, M. L. Jorgenson, J. R. Wingard, et W. B. Slayton, « Bone marrow sinusoidal endothelial cells undergo nonapoptotic cell death and are replaced by proliferating sinusoidal cells in situ to maintain the vascular niche following lethal irradiation », *Exp. Hematol.*, vol. 36, no 9, p. 1143-1156, 2008.

[7] W. H. Knospe, J. Blom, et W. H. Crosby, « Regeneration of locally irradiated bone marrow. I. Dose dependent, long-term changes in the rat, with particular emphasis upon vascular and stromal reaction », *Blood*, vol. 28, no 3, p. 398-415, 1966.

[8] J. Grassinger et S. K. Nilsson, « Methods to analyze the homing efficiency and spatial distribution of hematopoietic stem and progenitor cells and their relationship to the bone marrow endosteum and vascular endothelium », *Methods Mol. Biol. Clifton NJ*, vol. 750, p. 197-214, 2011.

[9] S. K. Nilsson, H. M. Johnston, et J. A. Coverdale, « Spatial localization of transplanted hemopoietic stem cells: inferences for the localization of stem cell niches », *Blood*, vol. 97, no 8, p. 2293-2299, 2001.

[10] C. Lo Celso et al., « Live-animal tracking of individual haematopoietic stem/progenitor cells in their niche », *Nature*, vol. 457, no 7225, p. 92-96, 2009.

[11] J. D. van Buul et al., « Migration of Human Hematopoietic Progenitor Cells Across Bone Marrow Endothelium Is Regulated by Vascular Endothelial Cadherin », *J. Immunol.*, vol. 168, no 2, p. 588-596, 2002.

Reviewer #1:

Summary:

While actin cytoskeletal remodeling is important in hematopoietic cell motility and migration, the precise mechanisms by which hematopoietic stem and progenitor cells (HSPCs) establish residency following homing to the bone marrow (BM) niche remain unclear. Two major protein families regulate actin nucleation and cellular motility in mammalian cells, (1) formin-related proteins and (2) WASP protein complexes. mDia1, mDia2, and mDia3 belong to a highly conserved family of diaphanous-related formins proteins that modulate a number of cell processes in addition to actin polymerization, including the modulation of microtubule networks, mRNA trafficking, serum response factor (SRF)-mediated gene expression, and cytokinesis. In this manuscript, Mei et. al. explore mechanisms of mDia2 (Diaph3) and, to a lesser extent, mDia1 (Diaph1)-mediated HSPC transmigration and engraftment into the HSPC-supportive bone marrow microenvironment using in vivo knockout strategies paired with genetic complementation to rescue HSPC deficiencies in a bone marrow transplantation (BMT) model system. In competitively transplanted recipients, loss of mDia2 (using a hematopoietic-specific Vav1-cre and Mx1-cre) results in distinct hematopoietic failure in non-competitive and competitive BMTs. Interestingly, impaired HSPC function is not due to a homing defect, but rather a failure of HSPCs to properly transmigrate from the peripheral blood to the BM parenchyma, and vice versa. This was confirmed using a combination of BM imaging following BMT, in vivo hematopoietic mobilization assays, and ex vivo migration assays of mDia2 knockout HSPCs. mDia1 knockout animals display a significant, but more mild engraftment phenotype (when compared to mDia2 knockouts). HSPC transmigration was not examined in mDia1 knockouts. Downstream of mDia2 activity, MAL-SRF signaling displayed functional deficiencies including a decrease in some canonical SRF targets and non-canonical beta2 integrins CD11a, CD11b, and CD18 targets. The authors confirmed enhancer binding activity in both CD11b and CD18 integrins using mutational enhancer analysis in a luciferase assay. Genetic complementation (overexpression) of SRF in mDia2 knockout HSPCs restored the observed engraftment phenotype. Moreover, the authors confirmed the function of SRF-dependent integrins by deleting CD11b and CD18 using CRISPR-Cas9 phenocopying the mDia2 knockout BMT mice. Genetic complementation (overexpression) of CD11b in mDia2 knockout HSPCs also partially restored the observed engraftment phenotype. The approach to examining the mechanism of mDia2 loss of function in HSPC engraftment in this manuscript is exhaustive and provides a significant springboard for future studies. The findings also provide significant insights into HSPCs migration and lodgement phenotypes, and should be of interest to the general readership of Nature Communications. The inclusion of data from mDia1 knockout mice is some cause for concern: (1) The phenotype in mDia1 mice are not examined in detail (compared to mDia2 knockout mice), (2) Analysis of mDia1/mDia2 double knockouts is incomplete, and (3) the mDia1 knockout model is a global knockout that may have additional niche-mediated effects on HSPCs. Moreover, the discussion points out a number of obvious questions that can be explained by mDia1 overlapping function. The inclusion of mDia1 in this story demands experimental answers to these questions (e.g. expansion of HSPCs phenotype in SRF knockouts, page 19 and partial SRF target gene dysregulation in mDia2 knockouts, page 20, and proper HSPC experimental phenotyping/MAL-SRF signaling investigation). As currently presented by the authors, the manuscript is incomplete. If the authors

choose to leave *mDia1* knockout model in the manuscript, they should address the questions raised in Major Points #5. Alternatively, this manuscript can focus on the *mDia2* story and address *mDia1* more completely in another manuscript.

We thank the reviewer's comments and suggestion. We also agree that it is more straightforward and focus on *mDia2* in the manuscript. As discussed above, we removed *mDia1* studies in the revised manuscript. Our point-by-point response to reviewer 1 is as below:

Major Points:

1. Can the authors explain the rationale for examining non-competitive BMT (Fig 1) at eight weeks post-transplantation (as opposed to standard 16-24 week time points in which homeostatic hematopoiesis reestablished)?

We appreciate the reviewer's comment. There was an error in presenting the data in old Figure 1. Old Figure 1A-E were data generated from 10-month old transplanted mice but not 2 months (these data are now placed in the supplemental Figure S2).

Following the reviewer's suggestion, we repeated the non-competitive BMT, and examined HSPCs 16 weeks post transplantation. As shown in Figure 1A-C in our revised manuscript, mice transplanted with *mDia2* deficient bone marrow mononuclear cells (BMMCs) displayed mild expansion of HSPCs in the bone marrow (with increased LT-HSC percentage and elevated HSPC cell numbers in general). This increase may reflect the compromised SRF activity in HSPCs, since SRF knockout mice show expansion of HSPCs in previous reports. In addition, we found that the LSK cells in the recipient mice transplanted with *mDia2* deficient BMMCs after 16-weeks lost stem cell quiescence demonstrated by the decreased percentage of cells at G0 phase but increased G1 phase (new Figure 1D). This phenotype became more significant at 10 months post transplantation (new supplemental Figure S2B). Therefore, we concluded that in the early stage of post-transplantation (4-month), depletion of *mDia2* induced a mild expansion of HSPCs in the bone marrow. However, with the loss of stem cell quiescence, the *mDia2* deficient HSPCs are exhausted when tested at the later stage of transplantation (10-month). We have included this in the revised discussion section.

Considering the dramatic engraftment phenotype in *Vav1-cre; mDia2^{fl/fl}* competitive BMT (Fig 2), does engraftment of HSPCs in *Mx1-cre; mDia2^{fl/fl}* in non-competitive BMT represent non-recombined HSPCs?

We performed a real-time PCR to confirm the recombination efficiency after transplantation. As shown below, *mDia2* mRNA level was dramatically decreased in mice transplanted with *mDia2^{fl/fl}* Mx-Cre BMMCs. Therefore, we believe that non-recombination does not play a role here. Instead, compensation by *mDia1* is likely the possible reason as demonstrated in our old Figure 7 with the DKO model. Since these data will not be presented in the revised manuscript, we discussed these findings instead.

Since *mDia2^{fl/fl}* Mx-Cre and *mDia2^{fl/fl}* Vav-Cre models show essentially the same phenotype, we removed non-competitive transplantation data of the *mDia2^{fl/fl}* Mx-Cre model.

2. The initial description and experimental usage of mouse models in this manuscript is confusing. Model generation: The methods and text do not adequately describe the generation of these models. Please provide source (catalog/vendor) information for the *Mx1-cre* and *Vav1-cre*. Is the *Vav1-cre* constitutively active during development or an inducible model? Experimental usage: What is the reasoning for utilizing both *Mx1-cre* and *Vav1-cre* models interchangeably when making hematopoietic comparisons (e.g. steady state Sup 1C, E vs BMT stress Fig 1A). What is the significance of comparing results from the two models (i.e. what new information does it tell the reader)? Why was *Mx1-cre* and *Vav1-cre* chosen for this

study? In some cases, data is left out entirely for comparisons between models (e.g. page 6, Mx1-cre steady state analysis).

We apologize for the lack of experimental details. These have been added in the revised methods section. Both Mx-Cre and Vav-Cre models allow reliable deletion of interested genes in the hematopoietic compartment. Vav-Cre mice express Cre under the control of mouse vav 1 oncogene promoter starting from embryonic day 12 (Genesis 34:251–256 (2002)). Mx-Cre model offers researchers the capability to induce hematopoietic knockout when mice reach adult. It is particularly useful with Mx-Cre model to deplete mDia2 after bone marrow reconstitution in old Figure 7A in our original submission. We utilized both Mx-Cre and Vav-Cre models to comprehensively investigate the role of mDia2 in HSPCs. To make the paper less confusing, we removed non-competitive transplantation data using Mx-Cre model since it is essentially the same as the Vav-Cre model. Old Figure 7 is removed as discussed above.

3. The observation that HSPCs fail to transmigrate into the in vivo bone marrow niche (Fig 2G-H) is critical piece of information for this manuscript. In addition to the in vitro migration assay, it directly implicates a lodgement phenotype. However, the images (Fig 2G) do not give the reader any magnitude of this observation. It is important to have a much more representative series of images to support the quantification (Fig 2H). This data can be amended as a supplementary Figure. Scale bars also needs to be added to all microscopic images. It would also be beneficial to test this hypothesis; does intrafemoral injection of mDia2 knockout HSPCs circumvent the observed engraftment phenotype?

We thank the reviewer's comments and suggestion. To give the readers a more apparent view, we highlighted the vessels with white dotted lines in Figure 2H. In addition, we provided representative images in supplementary Figure S4A to show the relative location (inside, outside and associated) of cells to the vessels. The scale bar was also included in new Figure 2H. In addition, following reviewer 3's suggestion, we performed a transplantation assay in non-irradiated recipient mice and observed the same defects of mDia2 deficient HSPCs in trans-endothelial migration in the bone marrow and spleen. Representative images are shown in new supplemental Figures S4B and S4C.

We also performed the intrafemoral injection of control or mDia2 knockout BMMCs with CD45.1 competitor BMMCs into lethally irradiated receipt mice. FACS analysis of peripheral blood was used to examine reconstitution. As show in supplemental Figure S4D in the revised manuscript, intrafemoral injection improved the competitive reconstitution ability of mDia2 deficient HSPCs in the recipient mice when compared to canonical retro-orbital injection route. This data indicates that intrafemoral injection can partially circumvent the engraftment defect in mDia2 deficient HSPCs.

4. No cell cycle differences were observed in steady state mDia2 knockout. Are there any cell cycle differences noted following hematopoietic recovery in non-competitive mDia2 BMT mice (Fig 1A-C)?

We performed cell cycle analysis in Vav-Cre model both at early and late stages post non-competitive transplantation. As shown in new Figure 1D and supplementary Figure S2B in our revised manuscript, a significant amount of mDia2^{fl/fl}Vav-Cre LSK cells entered cell cycle.

5. The interplay between mDia1 and mDia2 in this manuscript are very interesting, but raise a series of questions concerning their mutual (compensatory) and mutually exclusive functions that should be addressed experimentally.

We thank reviewer's comments about the role of mDia1. As discussed above, we removed mDia1 data in the revised manuscript following the reviewer's suggestion. Answers to the specific questions about mDia1 are provided below.

(1) mDia2 functions through the MAL-SRF signaling axis, but only alters a subset of canonical/integrin SRF targets. Are the remaining targets altered in mDia1 knockout animals?

In our previously published report (Blood, 124(5): 780-90), we performed a RNA-sequencing analysis in granulocytes from mDia1 knockout mice. As shown below, several SRF targets showed compromised expression, which is consistent with our conclusion in the old manuscript that mDia1 compensates mDia2's function.

(2) Is HSPC frequency and absolute numbers altered in under steady state and BMT conditions?

We characterized the absolute numbers of HSPCs both under steady state and bone marrow transplantation, which was shown in Figure S3 in our original submission. Briefly, there is no significant change of HSPCs in mDia1 KO mice at steady state. During transplantation stress, HSPCs declined in the recipient mice transplanted with mDia1 knockout BMMCs.

(3) What is the status of canonical/integrin SRF targets in the mDia1/mDia2 double knockout HSPCs?

We are currently performing RNA-sequencing analysis of cells from all four groups of HSPCs, including those from mDia1/mDia2 double knockout (DKO) mice. We expect to capture more down-regulated SRF targets in DKO HSPCs, which will be presented in our next manuscript.

(4) As noted by the authors, SRF knockout mice have a significant expansion of phenotypic HSPCs at steady state that was not observed in mDia2 knockouts. Do mDia1/mDia2 double knockouts display HSPC expansion under homeostatic conditions?

Thanks to the comments. Although we didn't capture an expansion phenotype in young mDia2 knockout mice at steady state, we did observe a slight expansion of HSPCs in the bone marrow in 2-year old mDia2^{fl/fl} Vav-Cre mice as shown below.

In addition, when performing non-competitive transplantation per the reviewer's suggestion in major point #1 (see above), we also observed a mild expansion of HSPCs at 4 months post-BMT as shown in new Figure 1A-C. These data are consistent with an attenuated SRF activity with mDia2 knockout.

Due to high mortality rate after birth in mDia1/mDia2 DKO mice, we performed HSPC analysis in neonates in Figure 7 of our original submission. DKO neonates didn't show any expansion but a decline of HSPCs in the bone marrow and spleen. These data will be presented in the next manuscript.

- (5) Does SRF rescue mDia1 single knockout or mDia1/mDia2 double knockout HSPCs?
- (6) Do mDia1 knockout mice display a transmigration defect similar to mDia2 knockouts (imaging, mobilization, Transwell assays)?
- (7) Is the transmigration phenotype more pronounced in mDia1/mDia2 double knockout HSPCs?

These are great questions. We will present these data in our next manuscript.

Minor Points:

1. *The intermittent inclusion of heterozygous deletion of mDia2 (e.g. Fig 2C) is distracting and does not add any value to the conclusions of the paper.*

We agree with the reviewer. We have removed the data of heterozygous deletion of mDia2 in new Figure 2.

2. *Figures use a combination of listed p-values and asterisks in bar graphs to indicate significance. Please correct for consistency.*

We thank the reviewer to point out this issue. Now we list all p-values as asterisks in our revised manuscript.

3. *Figure legends should have indications of which statistics were used to generate p-values.*

The information is included in the figure legend in the revised manuscript.

4. *Include significance for engraftment in Fig 2C.*

We now include p value in new Figure 2C. And we have confirmed that all the p values in mDia2^{fl/fl}Vav-Cre columns are less than 0.0001 when compared with control group.

5. *Include significance for engraftment in Fig 7A.*

We have removed original Figure 7A based on the reviewer's suggestion.

6. *An ANOVA analysis (repeated measures) may provide statistical power to Fig 7C.*

Thanks to this suggestion. We will perform an ANOVA analysis in our next manuscript. The original Figure 7 has been removed based on the reviewer's suggestion.

7. *The Methods sections lacks experimental detail in many sections (e.g. Clear explanations the mice/models used, how were the animals preconditioned for BMT, etc.). Please ensure transparency and clarity for the readers.*

We included detailed description of the mouse models when the mice were first introduced. We also include more information about BMT in the methods section.

Reviewer #2:

Homing and establishment of HS/PCs in the bone marrow is critical for establishment of hematopoiesis, but the molecular mechanisms involved are poorly understood.

In this study, Mei et al investigate the role of mDia1 and mDia2 formins in bone marrow engraftment of hematopoietic stem and progenitor cells (HSPCs). First they show that mDia2 KO cells show defective reestablishment of LSK and LT/ST-HSC populations, with LSK KO cells showing loss of quiescence. Similar but weaker phenotypes were seen with mDia1 KO., which appears to synergies with mDia2. They show that mDia2KO did not affect homing of the HSPC to the bone marrow vasculature but led to impaired transendothelial migration and niche engraftment, and defective mobilization by GCSF. This was associated with inactivation of the MAL transcriptional coactivator, which became refractory to serum stimulation, and reduced expression of MAL-SRF target genes. The authors provide evidence that Itgb2 and ItgaM are direct targets for SRF. They go on to show that retroviral transduction of SRF restores classic SRF target gene expression as well as integrin expression, adhesion, and engraftment.

Next they use CRISP/Cas of *ItgaM* or *Itgb2* to show that these integrins are required for recruitment, their combined deletion being synergistic. Ectopic expression of *ItgaM* can partially rescue the *mDia2KO* phenotype.

EVALUATION

The paper nicely shows that MAL-SRF signaling is downstream of *mDia2* in HSCP engraftment, and that integrin expression is an important aspect of this with *ItgaM* and *Itgb2* playing important roles. It also shows redundancy with *mDia1*. What it does not do is convincingly show that MAL-SRF signaling activates the integrins directly, as opposed to indirect effects arising through MAL-SRF's well known role in regulating expression of multiple cytoskeletal genes. The paper is marred by an unconvincing analysis of the integrin regulatory sequences and particularly poor presentation of this section.

We thank the reviewer's comments. Please see below for our point-by-point response:

Major issues:

1. Ectopic expression of *ItgaM* can partially rescue the *mDia2KO* phenotype, but *Itgb2* is not tested, alone or in combination with *ItgaM*. What happens with *Itgb2* ± *ItgaM*? Moreover, they authors do not test *ItgaL* either. These experiments should be done and the results shown.

We thank the reviewer's comment. We performed the rescue experiment with *Itgb2*. However, overexpression of *Itgb2* failed to rescue the defects (new supplemental Figure S7G). We hypothesized that the expression of *Itgb2* must be precisely regulated in that retrovirus-mediated over-expression of *Itgb2* could be toxic to HSPCs. This could also be the reason that *Itgb2+ItgaM* did not rescue *mDia2 KO* phenotype as well. These are discussed in the revised manuscript.

We did not test *ItgaL* for the rescue experiments in the manuscript. First, *ItgaL* was less downregulated compared to *ItgaM* and *Itgb2* when screened by real-time PCR in Figure 3G. Second, when searching for the ChIP-seq binding peak dataset of SRF in the genome scale, we did not identify a potential SRF binding element in *ItgaL*. Therefore, we focused on *ItgaM* and *Itgb2* in this study. This is explained in the revised manuscript.

2. A critical point of the authors' model is that *mDia2KO* cells should be defective in recruitment of MAL to SRF binding sites (ANY SRF binding sites that recruitment MAL). This can be demonstrated by ChIP, but no data are provided. This is essential to validate the proposed integrin SRF binding sites, and for canonical target genes such as *Acta2* or SRF itself.

We agree with the reviewer and performed the experiments. We performed chromatin IP with antibodies against SRF, MAL followed by qRT-PCR of *ItgaM*, *Itgb2*, and *Acta2*. In new Figure 4H and S6B, we show enriched binding of SRF and MAL on *ItgaM*, *Itgb2* as well as *Acta2* gene locus upon serum stimulation.

3. The SRF binding analysis is not adequately presented and does not cite the appropriate literature.

The SRF binding sites were predicted through an online program (TFBIND, developed by Tatsuhiko TSUNODA, <http://tfbind.hgc.jp/>). We included additional reference in the revised manuscript. (T.Tsunoda, and T.Takagi. Estimating Transcription Factor Bindability on DNA. *Bioinformatics*, Vol.15, No.7/8, pp.622-630, 1999).

Other Points:

1. Order of presentation. Two points:

- The presentation of the *Dia1* data occurs in Figure S3(p8), then in Figure 7 (p17), and as a result the Figure 7 data reads as an afterthought. Should present the *mDia1* combination with *mDia2* before dealing with mechanism.

We agree with the reviewer. Based on the suggestions from Reviewer #1, we have removed all mDia1 KO and mDia1/mDia2 DKO related data in the revised manuscript to focus the paper on mDia2.

- The Integrin deletion and rescue data – strong data that show that it is integrin expression is limiting upon mDiaKO – could be presented before the data showing that they are SRF targets.

We thank the reviewer's suggestion. However, we still feel it is more readable and logic to present mDia2-SRF-integrin regulatory cascade before rescue experiments in vivo. We hope the reviewer will agree with us.

2. Absolute cell counts should be displayed in Figure 1D

We have updated the results in new Figure 1C.

3. The transwell assay in Figure 2F could be performed on a monolayer of endothelial cells as well, which would better mimic the in vivo context and confirm whether the transendothelial migration of KO cells is defective in vitro.

We thank the reviewer's suggestion. In the revised manuscript, we utilized mouse endothelial cells seeded on the trans-well and loaded the lineage negative cells on the upper chamber. We subsequently collected the cells from the lower chamber and performed colony-forming assay with semisolid Methylcellulose media. As shown in new Figure 2G, mDia2 knockout cells exhibited dramatically compromised migration ability through the endothelial cell monolayer.

4. Figure 3E/F. The MAL and SRF and gene expression data is not convincing.
3E -The localization figure needs quantitation, it is only shown as images of representative cells, but it should also be quantified over 50-100 cells in Figure 3E, with markers for the cytoplasm.

We repeated the immunofluorescent staining with a scaffolding component of the nuclear envelope lamin B1 (new supplementary figure S5A) and quantified the location of MAL in new Figure 3E. The cytoplasmic markers turned out to be difficult to use in HSPCs given the small amount of cytoplasm in these cells.

3F - some these genes are MAL-controlled and others by the Elk1 transcription factor but it looks like there are Elk targets affected by mDia2KO (eg Egr1) and MAL targets that are not (eg Vcl, Tgln). What is going on? What happens if the serum-stimulated cells in 3E are analyzed by RT-PCR?

We thank the reviewer's comment. We believe that the cell context-dependent gene regulation by the transcriptional factors could play a role in this case. Additionally, according to the published data (*Dev Cell*. 2014 Nov 10;31(3):332-344), except KRT17, the 7 targets of MRTF, including MRTF-A (also known as MAL), in their ChIP dataset including Acta2, Flna, FHL2, Myh9, Actb, Actg1 and SRF showed significant decrease in our results. The reference data are shown below for the reviewer's convenience.

[Redacted]

[Redacted]

(*Genes Dev.* 2014.28(9):943-58. Figure 2E) (*Genes Dev.* 2014.28(9):943-58. Supplemental Table S5)

Cytoskeleton changes usually modulate SRF activity through MRTF, which have been extensively studied both in mouse and human. (*Genes & Dev.* 2014. 28: 943-958. *Science*,2013. 340: 6134. 864-867.)

Both Egr1 and Egr3 are also binding targets of MRTF, as shown in Figure S5B in Mol Cell. 2016. 64(6):1048-1061 (see data below). Consistently, these two genes showed a decrease trend in our mDia2 cKO HSPC.

[Redacted]

(Mol Cell. 2016. 64(6):1048-1061. Figure S5B. WT column with or without 15% FCS)

Meanwhile, in Figure 6A of Gualdrini et al. (Molecular Cell, 2016, 64, 1048–1061, see below), 15% FCS dramatically stimulated the SRF binding compared with 0.3% FCS, indicating the binding activity is significantly enhanced by serum stimulation, this is also observed by another group through CHIP-sequencing (Genes Dev. 2014.28(9):943-58.). The data are presented below for the reviewer's convenience.

[Redacted]

[Redacted]

*(Mol Cell. 2016. 64(6):1048-1061.
2014.28(9):943-58.
Figure 6A, WT with or without 15% FCS)
Acta2 locus)*

*(Genes Dev.
Figure 1B ChIP-seq peaks on*

We stimulated the cells with 10% FBS for 30 minutes followed by qRT-PCR as the reviewer suggested. As shown in supplementary Figure S5B and C in our revised manuscript, serum treatment triggered a dramatic induction of *ItgaM*, *Itgb2* as well as selected SRF target genes (*Acta2*, *KRT17* and *Flna*).

3F – relative expression data do not agree with Costello et al. which estimated relative WT/KO as Itgb2: 1.14; Itgal: 1.01; Itgam: 0.75. Please comment.

We thank the reviewer's comment. Costello et al. collected LSK cells from E14.5, whereas we used adult c-Kit+ HSPCs. The difference may be due to the variation of developmental stages.

Figure 3K – adhesion should also be performed on an ICAM substrate, which is one of the main physiological substrate of integrins alphaM/beta2.

We performed the experiment as the reviewer suggested. ICAM-1 coated coverslips were prepared and used for the adhesion assay. As shown in new Figure 3L, mDia2^{fl/fl} Vav-cre HSPCs showed significant defects in adhesion to ICAM-1 substrate.

5. Figure 4 is not acceptable, and the authors appear to be unaware of the classical SRF literature.
- Figure 4A. The SRE sequence consensus is CC(A/T)6GG, which should be referenced properly. There is no good fit to this consensus in either of these two sequences. What do the arrows indicate?

We included references of SRE in the revised manuscript. We agree with the reviewer that SRE on *Itgb2* intronic region does not perfectly match the classical SRE consensus sequence. However, the SRE on *ItgaM* does contain CC (A/T rich) GG motif [cc(ttgaga)gg], which was viewed as variants of the motif (Miano et al, *Am J Physiol, Cell Physiol*, 2007, 292(1) C70-81). Furthermore, recent studies using SRF ChIP-seq revealed SREs that do not contain CC(A/T)6GG consensus sequence, such as SP-1, ETS, and GFY as shown below (*Genes & Dev.* 2014. 28: 943-958). We discussed these points in the revised manuscript.

[Redacted]

(*Genes & Dev.* 2014. 28: 943-958, Figure 2F)

The arrows indicated the predicted direction of SRE. We removed the arrows in our revised manuscript.

- Figure 4C/D. These are not consistent. Why does SRF cause a 3 fold effect on *ITGAM2+* in C, but an 8fold effect in D?

We noticed the difference between these data. These experiments were done separately, therefore, variation could play a role here. In addition, the transfection efficiencies were also different in which we transfected less amount of Renilla pRL-TK in Figure 4D (0.5 ng/well in 4D and 1 ng in 4C) since 4D involves increasing amount of SRF. We apologize for the confusion and have included detailed transfection information in the method section.

- Figure 4F Assuming that the CCTTGAGAGG is a consensus, there is NO published literature supporting the notion that the C at position +6 downstream would have any effect on SRF binding – why was this mutant chosen? Where are two sites, and what is the evidence that they increase DNA binding activity?

We found this site incidentally. This T site was spontaneously mutated to C in a PCR reaction in our template plasmid. Surprisingly, overexpression of this mutant hardly induce the luciferase expression. When this T/C mutation was corrected to T, the luciferase expression was restored. We included discussion in the revised manuscript.

Need remove site 1 in main figure.

This is removed in the revised manuscript.

- Figure 4G –Figure 4A bottom should be modified to match this. Where are the proposed SRE elements in this sequence, and why were the point mutations chosen? And what is a “G/C to T” mutation?

As discussed above, we predicted the SRE elements using an online program (TFBIND) and found that the site is not the consensus SRE sequence. Since C/G is critical for the consensus sequence, we mutated C and G in this non-canonical SRE site (TTTAACATACAAGGCCAT) to T (TTTAATATACAATTTTAT), named as “G/C to T” mutation. Figure 4G showed that this “G/C to T” mutation abolished luciferase activity induced by SRF. To match Figure 4A and make it clearer to the readers, we delete site 2 annotation in Figure 4G in our revised manuscript. G/C to T mutation is also explained in the revised manuscript.

- Figure 4H. This experiment is not explained. What is the significance of the +10% FBS? SRF is meant to bind constitutively to its targets. Surely the authors should see a signal in the “-“ lane too? A control with isotype-matched irrelevant antibody should be shown. What is the ChIP signal on a bona fide SRF target such as *Acta2* or SRF itself? The authors argue that *mDia2* controls MAL to regulate these integrins, so should also show a MAL ChIP. Finally a critical point in the authors’ model is that *mDia* cells should show a defect in MAL recruitment to SRF sites. This should be done.

Thanks to the reviewer’s comments. We partially answered these points in Other points #4 regarding Figure 3F raised by the reviewer. Previous studies by Esnault et al (Genes & Dev. 2014. 28: 943-958, shown below) revealed a SRF transcriptional regulation pathway in that many serum inducible genes are barely bound by SRF in resting cells. The SRF/MRTF complex cooperatively excludes nucleosomes at the SREs upon serum stimulation.

[Redacted]

(Genes & Dev. 2014. 28: 943-958. Figure 3I.)

Following the reviewer’s suggestion, we performed both anti-SRF and anti-MAL ChIP assay using anti-GFP as a negative control. As we mentioned in our response to Major issue #2 from the reviewer, we captured specific enrichment of SRF and MAL binding on *ItgaM*, *Itgb2* as well as *Acta2* gene locus upon serum stimulation, which is absent with GFP antibody IP. Loss of *mDia2* abolished these binding activities.

- Figure 4I. The CRISPR/Cas experiment is not adequately explained in the text or figure legends. It appears that there are multiple edits in the population (Figure 4D), yet a single population is shown for CD11B analysis - explain. Moreover, the effect of the deletion should be evaluated by qRT-PCR on *ITGAM2*, not by CD11 MFI.

We thank the reviewer’s comment, but we are not sure that we understand the reviewer’s question. Figure 4D is the luciferase assay showing dose-dependent response by SRF overexpression and has nothing to do with editing. Following the reviewer’s suggestion, we sorted the GFP/tRFP657 double positive LSK cells from sgLuc2p, sglItgaM intron2-T7, and sglItgaM intron2-T26 transduced cells in the transplant recipients and quantified the mRNA levels of *ItgaM* expression. As shown in new Figure 4I (right panel), genomic editing the SRE in *ItgaM* intronic region significantly reduced the transcripts of *ItgaM*.

5. The competitive BMT experiment in Fig S3D made use of GFP labelled KO cells, tracking them 2 months post-transplantation; are cells infected with GFP expressing viral particles, or is GFP transgenically expressed? if cells were transduced, to control for the fact that the cells could have lost the GFP during this time, the reverse labelling experiment should be performed, with unlabelled KO cells and GFP labelled WT cells; alternatively, a CD45.1/CD45.2 system could be used.

The *mDia1* knockout mice express GFP under its own promoter in germline. We obtained the mice from Dr. Arthur Alberts (Van Andel Research Institute) (Peng et al. 2007 *Cancer Res* 67:7565). The *mDia1* knock-out allele contains EGFP that replaces part of exons 2 and 6 and all of exons 3-5 of the *Diap1* locus. This abolishes endogenous gene function and results in the expression of EGFP (information from The Jackson laboratory, Stock No:030411). Due to high endogenous levels of *mDia1* in mouse hematopoietic system, almost all the circulating mononuclear cells express GFP. Therefore, we took this advantage to

monitor mDia1 knockout cells in vivo. Alternatively, we also used the CD45.1/CD45.2 system, which was shown in supplementary figure 3E in our original submission.

Nevertheless, as we discussed in the beginning, we removed all the data related to mDia1 KO and mDia1/mDia2 DKO mice in our revised submission per reviewer #1's suggestion.

6. *Figure S3E The second competitive BMT here has not been initiated with a 1:1 mix, but with only 20% of KO cells, which seems to be the ratio at which they are found 6 months after the first cBMT; please clarify in the legend.*

Thanks for the reviewer's reminding. We will clarify this point in our next manuscript. Now we have removed the figure per reviewer #1's suggested.

7. *Figures 2G&H It would have been good to have both WT & KO labelled with different dyes & injected into the same recipient. Limited information is presented - just two cells for the visual representation. The Figure needs to include details on numbers of cells counted per field, how many cells are there in 5-6 fields?*

Please also refer to our response to reviewer #1's comment 3. When trying to label WT&KO cells with two different fluorescent dyes, we encountered high levels of background that prevented meaningful characterization. We now included the cell numbers in the revised figure legend. Additionally, per reviewers #1 and 3's suggestion, we performed more in vivo imaging in bone marrow and spleen in a short period after non-irradiation transplantation (new supplementary figure S4B and S4C).

8. General Figure issues

- *Statistical data (p-values and n) are missing in some panels, they should be added to all figure panels, and the statistical test used should be stated in the legends.*

- *for the Kaplan-Meier survival curves, median survival for each condition (including control) should be clearly indicated in Figures 1F, 1G, 1H, 5J, 6G, 7B, 7H.*

These have been corrected in the revised manuscript.

- *the y-axis should not be discontinued in graphs, it artificially amplifies small differences between conditions: please correct accordingly Figures 1C, 4J, S6A, S6E.*

We corrected most of the figures. However, some figures, such as new supplemental Figure S1D, it is reasonable to use discontinued Y axis to illustrate the values of some lineages. Otherwise, they can hardly be seen in the figure.

- *red and blue columns have been swapped in Figure 3H, please correct.*

We thank the reviewer for pointing out the error. This is corrected.

- *the legends of the x-axis (0 to 7 months post-cBMT) in Figure 7A should be shifted towards the left, consistent with plpC injections one month after transplantation*

Based on reviewer 1's suggestion, original Figure 7 for mDia1/mDia2 double knockout mice was removed.

- *Figure S5, please specify in the legend what the blue shading indicates*

This information is included.

9. *Results p10: 'mDia2 regulates HSPC engraftment through SRF signaling'. This statement is too strong – these results shows that loss of mDia2 in HSPC results in defective F-actin, cell polarity, adhesion, MAL-SRF pathway and integrin beta2 expression, but not that this is cause of the engraftment defect. Please rephrase.*

We agree with the reviewer. The sentence is rephrased.

10. P10§3 should read "... reported to affect cell functions through INactivation of the MAL-SRF pathway in cell based assays"

reference 40 is not about MAL. Replace.

Numerous typos – correct please.

Refs 27 and 66 are the same!

The comment in the discussion that transcription factors binding sites are located 5' of the TSS on promoter regions, and that an intronic binding site is unusual, is simply not true. Remove.

Thanks for the reviewer's careful reading. We have corrected them in the revised manuscript.

Reviewer #3:

In their manuscript, Mei et al. demonstrate that mDia1 and 2 -proteins involved in the regulation of actin-are involved in HSPCs homing, engraftment, and repopulation potential of lethally-irradiated mice. Whereas the transendothelial migration of leukocytes has been studied in detail before, this is not the case for HSPCs. Therefore, the manuscript substantially adds to the field. However, how mDia deletion impacts transplanted HSPCs is not so clear. Below are some points aimed at sharing clarity in this regard and improve the manuscript.

1. Endothelial cells lining BM sinusoids have been identified as critical for HSCs homing and engraftment [1]–[3]. However, it is well known that irradiation damages the niche, in particular sinusoidal endothelial cells [3]–[7]. Therefore it is unclear to what extent the findings reflect physiological migration rather than BM engraftment and reconstitution after myeloablation. Protocol in non-irradiated mice have been described [8]. Although the authors state that "the bone marrow vasculature remained intact", Fig 2G-H shows disruption of BM vessels stained with CD31 which, together with increased vessel permeability, makes difficult to localize cells inside/outside of vessels. To address this, HSPC transmigration through vessels could be analyzed through intravital imaging in non-irradiated mice <15h after transplantation [8] for confirmation.

We thank the reviewer's comments. We performed non-irradiation transplantation as the reviewer suggested in the revised manuscript. 5 hours after retro orbital injection, images were taken from the bone marrow and spleen. The relative location of cells to the vessels in the indicated organ were visualized and quantified. As shown in supplementary figure S4B and S4C, mDia2 knockout cells exhibited dramatically attenuated transmigration ability out of the vessels.

2. While it is clear that mDia1 and mDia2 have a role in the maintenance of transplanted HSCs and their repopulation potential, the mechanism seems unclear. mDia deletion could lead to impairment in transendothelial migration, or in the maintenance of HSPCs inside of their niche by altering their adhesion to specific ECM protein (like fibronectin as you describe) or specific stromal cells, or both. For instance, the effects of mDia on HSC might be triggered by defective adhesion (rather than/besides transmigration impairment), which could disrupt HSC quiescence leading to exhaustion. The different possibilities and the most important mechanism should be better demonstrated and explained. It would be interesting to assess if Itg B2 re-expression can restore the proliferation pattern in transplanted HSCs mDia2-deficient HSCs.

We agree with the reviewer that the mDia2 loss may contribute to both impairment of trans-endothelial migration mainly in competitive transplantation (point 1#) and altering adhesion to niches leading to loss of HSPC quiescence and exhaustion in non-competitive transplantation (point 2#). We include detailed discussion incorporating our experimental evidence to support both claims in the revised manuscript.

For point #1, we uncovered the decreased expression of *ItgaM* and *Itgb2* in mDia2 HSPCs (Figure 3), which is mediated by compromised SRF activity (Figure 4/Figure 5). Although *ItgaM/Itgb2* complex have been recognized as key molecules in neutrophil trans-endothelial migration, their roles in HSPCs are still

controversial and not elucidative. We have confirmed the trans-endothelial migration defects of mDia2 deficient HSPCs by intravital imaging both at irradiation and non-irradiation status (Figure 4H/I and supplementary figure S4B/C). More importantly, re-expression of *ItgaM* significantly rescued the engraftment defect of mDia2 knockout HSPCs in competitive transplantation (Figure 6H).

For point #2, we found decreased adhesion of mDia2 knockout HSPCs to the ECM proteins in vitro (Figure 3K and 3L). We analyzed the non-competitive transplants both at early (4 months, Figure 1A-D) and late (10 months, supplementary figure 2A-E) period after transplantation. We found that at the early stage, mDia2 knockout HSPCs were slightly expanded in the bone marrow with loss of HSC quiescence. This is consistent with previous studies that ICAM-1 deficiency in the bone marrow niche impairs HSC quiescence (Stem Cell Reports. 2018 Jul 10; 11(1): 258–273). The HSC quiescence defects continue to the late stage that eventually led to exhaustion and reduction in HSPC populations.

To test if *Itgb2* re-expression restored the proliferation pattern in transplanted mDia2-deficient HSCs, we examined the recipient mice 6 months after a secondary transplantation. As shown below, *ItgaM* or *Itgb2* re-expression in mDia2 knockout cells restored the cell cycle profile, HSPCs composition, and absolute cell number of the LK population in the bone marrow. However, simultaneously expression of *ItgaM* and *Itgb2* led to the reduced cell number of LSK and LK populations, which again suggests that *Itgb2* level needs to be fine-tuned as we discussed above in reviewer #2's Major issue 1. We included new data of *Itgb2* overexpression in supplemental Figure S7G and discussion of these findings.

3. Fig 5: It would be better to show *Itg* surface expression by flow cytometry (rather than qpcr). Coculture assays of mDia2-deficient HSCs with endothelial cells (and not only fibronectin) with/without *Itg B2* inhibition would help substantiate the contention that mDia2 regulates trans-endothelial migration through *Itg B2*.

We thank the reviewer's comments and suggestions. The surface expression of beta2 integrins in mDia2 KO HSPCs assayed by flow cytometry was presented in original submission Figure 5B (now as Figure 5A in the revised version). Trans-endothelial migration was also performed in the presence of mouse endothelial cells as shown in new Figure 2G following the reviewer's suggestion.

4. The introduction can be improved. The authors state that the bone marrow repopulation is a two-step process, with homing occurring within 24h, followed by a long-term engraftment requiring lodgment to specific niches. However, the current bibliography indicates that immediately (1 hour) following a HSPC transplantation in lethally-irradiated mice the majority of donor cells locate in the central marrow region. This is followed by a rapid redistribution of the cells within 5h, resulting in a preferential seeding of LT-HSC-enriched cells closest to the bone surface, whereas more mature cells progressively engraft further away from bone [9], [10]. The bibliographic reference that is supposed to support the statement (ref 4 of the manuscript) is a review about the development of CXCR4 and VLA-4 inhibitors and how they may improve the utility and convenience of peripheral blood stem cell transplantation, which seems disconnected. Even if scarce, some studies regarding HSPC transmigration have been reported and should be presented in the introduction.

We thank the reviewer's comments. We included the information in the revised introduction and the references. We also included reference 11 the reviewer provided in the discussion.

5. Fig 3H: Does the frequency of cells expressing these integrins change upon mDia2 deletion? It has

been reported that VE-cadherin regulates hHSPCs trans endothelial migration [11]. Does VE-cadherin cell surface expression change in mDia2^{fl/fl} Vav-Cre mice?

As we shown in new Figure 3H, the frequency of cells with relative high integrin expression in the subpopulation of HSPCs are decreased. We performed the flow cytometry assay to analyze VE-Cadherin expression. As shown below, we found a subtle but not significant decrease of VE-Cadherin expression in LSK cells, but a significant increase of expression in LS cell population. The reasons for the differences in these two populations in the expression of VE-cadherin could be complicated and we prefer not to include the data in the current study. In addition, reference 11 the reviewer provided was VE-cadherin's role in endothelial cells. It's role in HSPCs was not known.

6. Fig 5A,F might be better placed in supplementary data for more streamlined presentation of the main data.

We agree with the reviewer. These data regarding the infection efficiency were moved to supplementary data.

7. Describing more clearly the relationships between CD11a/CD11b/CD18 and Itgb2 in the manuscript would help the reader. Few typos should be corrected throughout the manuscript. This sentence needs to be rephrased: "how transendothelial migration of the HSPCs in vivo are rarely investigated".

We thank the reviewer's comment. These were corrected accordingly.

[1] S. T. Avecilla et al., « Chemokine-mediated interaction of hematopoietic progenitors with the bone marrow vascular niche is required for thrombopoiesis », Nat. Med., vol. 10, no 1, p. 64-71, 2004.

[2] J. M. Butler et al., « Endothelial cells are essential for the self-renewal and repopulation of Notch-dependent hematopoietic stem cells », Cell Stem Cell, vol. 6, no 3, p. 251-264, 2010.

[3] A. T. Hooper et al., « Engraftment and reconstitution of hematopoiesis is dependent on VEGFR2-mediated regeneration of sinusoidal endothelial cells », Cell Stem Cell, vol. 4, no 3, p. 263-274, 2009.

[4] B. O. Zhou, L. Ding, et S. J. Morrison, « Hematopoietic stem and progenitor cells regulate the regeneration of their niche by secreting Angiopoietin-1 », eLife, vol. 4, p. e05521, 2015.

[5] H.-G. Kopp et al., « Thrombospondins deployed by thrombopoietic cells determine angiogenic switch and extent of revascularization », J. Clin. Invest., vol. 116, no 12, p. 3277-3291, 2006.

[6] X.-M. Li, Z. Hu, M. L. Jorgenson, J. R. Wingard, et W. B. Slayton, « Bone marrow sinusoidal endothelial cells undergo nonapoptotic cell death and are replaced by proliferating sinusoidal cells in situ to maintain the vascular niche following lethal irradiation », Exp. Hematol., vol. 36, no 9, p. 1143-1156, 2008.

[7] W. H. Knospe, J. Blom, et W. H. Crosby, « Regeneration of locally irradiated bone marrow. I. Dose dependent, long-term changes in the rat, with particular emphasis upon vascular and stromal reaction », Blood, vol. 28, no 3, p. 398-415, 1966.

[8] J. Grassinger et S. K. Nilsson, « Methods to analyze the homing efficiency and spatial distribution of hematopoietic stem and progenitor cells and their relationship to the bone marrow endosteum and vascular endothelium », Methods Mol. Biol. Clifton NJ, vol. 750, p. 197-214, 2011.

[9] S. K. Nilsson, H. M. Johnston, et J. A. Coverdale, « Spatial localization of transplanted hemopoietic stem cells: inferences for the localization of stem cell niches », Blood, vol. 97, no 8, p. 2293-2299, 2001.

[10] C. Lo Celso et al., « Live-animal tracking of individual haematopoietic stem/progenitor cells in their niche », *Nature*, vol. 457, no 7225, p. 92-96, 2009.

[11] J. D. van Buul et al., « Migration of Human Hematopoietic Progenitor Cells Across Bone Marrow Endothelium Is Regulated by Vascular Endothelial Cadherin », *J. Immunol.*, vol. 168, no 2, p. 588-596, 2002.

We thank the reviewer's reference. Some of these are included in the revised manuscript.

Reviewers' comments:

Reviewer #1 (Remarks to the Author):

Mei et al are submitting their revised manuscript, now entitled "Diaphanous-related formin mDia2 regulates beta2 integrins to control hematopoietic stem and progenitor cell engraftment". The authors have chosen to remove the incomplete dataset describing the mDia1 knockout and mDia1/mDia2 knockouts, and focus the manuscript on mDia2 function in hematopoietic stem and progenitor engraftment (therefore excluding the response to Major Point 5 from the original review). In support of the mDia2 knockout phenotype, the authors have provided additional experimental data to address the Major concerns cited, including: (1) Analysis of (New Figure 1) non-competitive Vav1-cre; mDia2fl/fl BMT 16 weeks post-transplantation; HSPCs now demonstrate an expansion of phenotypic HSPCs and an increase in cell cycling. This is more reflective of the previously described SRF knockout mice. Previously mislabeled 10 month post-transplantation data has been moved to Supplemental Figure 2B. (2) Addition of representative images of in vivo transmigration with improved delineation of the vasculature and (3) intrafemoral HSPC injections that partially circumvent the lodgment phenotype. In summary, the authors have adequately addressed most concerns and have significantly strengthened the manuscript. However, there are Minor points that the authors can address via changes to the figures/text to improve the manuscript.

Minor Points:

1. Page 5, line 108-111; please immunophenotypically define all populations (i.e. MPPs, LT-HSCs, ST-HSCs).
2. Page 6, line 122; "were transplanted non-competitively into lethally irradiated CD45.1+ mice.
3. While not necessary for all cell cycle analysis, can the authors include a representative flow plot in the main figures (e.g. for Figure 1D)?
4. Figure 2C; please fully define "Gran" and "MO" cell types.
5. Page 7, line 155-157; "Importantly, the absence of mDia2 deficient cells in LSK, LK, LT-HSC, and ST-HSC, and multipotent progenitor (MPP) populations (Figure 2B-C...). I do not see this data in Figure 2.
6. Vav1-cre; mDia2fl/+ heterozygous knockout data was removed from new Figure 2, but is still in Sup. Figure 1D and F. Please remove.
7. Supplemental Figure 2C-E; please include long-term competitive engraftment data, if available. In addition, it is not clear why the non-competitive Mx1-cre; mDia2 knockout data was removed. I believe the Mx1-cre; mDia2 knockout data is included to confirm a postnatal (and not developmental) HSPC phenotype. Moreover, please detail in the discussion what information the Mx1-cre data tells the reader in this system (i.e. confirming a postnatal and not a developmental phenotype).
8. Is the hCD4+ labeling (Y-axis) of Figure 5A correct? Please address.

Reviewer #2 (Remarks to the Author):

See attached annotated rebuttal letter

Reviewer #3 (Remarks to the Author):

The authors have adequately addressed the comments from the reviewers and this very interesting study seems now acceptable for publication

Reviewer #1:

Summary:

Mei et al are submitting their revised manuscript, now entitled “Diaphanous-related formin mDia2 regulates beta2 integrins to control hematopoietic stem and progenitor cell engraftment”. The authors have chosen to remove the incomplete dataset describing the mDia1 knockout and mDia1/mDia2 knockouts, and focus the manuscript on mDia2 function in hematopoietic stem and progenitor engraftment (therefore excluding the response to Major Point 5 from the original review). In support of the mDia2 knockout phenotype, the authors have provided additional experimental data to address the Major concerns cited, including: (1) Analysis of (New Figure 1) non-competitive Vav1-cre; mDia2^{fl/fl} BMT 16 weeks post-transplantation; HSPCs now demonstrate an expansion of phenotypic HSPCs and an increase in cell cycling. This is more reflective of the previously described SRF knockout mice. Previously mislabeled 10 month post-transplantation data has been moved to Supplemental Figure 2B. (2) Addition of representative images of in vivo transmigration with improved delineation of the vasculature and (3) intrafemoral HSPC injections that partially circumvent the lodgment phenotype. In summary, the authors have adequately addressed most concerns and have significantly strengthened the manuscript. However, there are Minor points that the authors can address via changes to the figures/text to improve the manuscript.

We thank the reviewer’s comments on our manuscript.

Minor Points:

1. Page 5, line 108-111; please immunophenotypically define all populations (i.e. MPPs, LT-HSCs, ST-HSCs).

We included detailed information about these populations in the revised main text.

2. Page 6, line 122; “were transplanted non-competitively into lethally irradiated CD45.1+ mice.

We believe that the reviewer would like us to include “non-competitively” in the sentence, we agree.

3. While not necessary for all cell cycle analysis, can the authors include a representative flow plot in the main figures (e.g. for Figure 1D)?

We now include a representative flow plot in new Figure 1D.

4. Figure 2C; please fully define “Gran” and “MO” cell types.

We now include the information in the revised figure legend.

5. Page 7, line 155-157; *“Importantly, the absence of mDia2 deficient cells in LSK, LK, LT-HSC, and ST-HSC, and multipotent progenitor (MPP) populations (Figure 2B-C...). I do not see this data in Figure 2.*

We thank the reviewer’s careful reading of our manuscript. This is a mislabel when preparing the revised figures. The HSPC populations should be at the bottom of Figure 2C. Now we have corrected these mislabels.

6. *Vav1-cre; mDia2fl/+ heterozygous knockout data was removed from new Figure 2, but is still in Sup. Figure 1D and F. Please remove.*

We now removed these data.

7. *Supplemental Figure 2C-E; please include long-term competitive engraftment data, if available. In addition, it is not clear why the non-competitive Mx1-cre; mDia2 knockout data was removed. I believe the Mx1-cre; mDia2 knockout data is included to confirm a postnatal (and not developmental) HSPC phenotype. Moreover, please detail in the discussion what information the Mx1-cre data tells the reader in this system (i.e. confirming a postnatal and not a developmental phenotype).*

For Supplemental Figure 2C-E, we unfortunately did not continue with for the long-term competitive engraftment experiment since their phenotypes are similar to the Vav-Cre model. For the non-competitive data about Mx1-Cre model, we removed due to the same reason that their phenotypes are similar to the Vav-Cre model. Now we included them in the revised manuscript.

8. *Is the hCD4+ labeling (Y-axis) of Figure 5A correct? Please address.*

We corrected the labels. We examined the expression of these integrins only in the transduced cells that are human CD4 (hCD4) positive since hCD4 is co-expressed in the viral vector.

Reviewer #2:

Major issues:

1. *Ectopic expression of ItgaM can partially rescue the mDia2KO phenotype, but Itgb2 is not tested, alone or in combination with ItgaM. What happens with Itgb2 ± ItgaM? Moreover, they authors do not test ItgaL either. These experiments should be done and the results shown.*

We thank the reviewer’s comment. We performed the rescue experiment with *Itgb2*. However, overexpression of *Itgb2* failed to rescue the defects (new supplemental Figure S7G). We hypothesized that the expression of *Itgb2* must be precisely regulated in that retrovirus-mediated over-expression of *Itgb2* could be toxic to HSPCs. This could also be the reason that *Itgb2+ItgaM* did not rescue mDia2 KO phenotype as well. These are discussed in the revised manuscript. We did not test *ItgaL* for the rescue experiments in the manuscript. First, *ItgaL* was less downregulated compared to *ItgaM* and *Itgb2* when

screened by real-time PCR in Figure 3G. Second, when searching for the ChIP-seq binding peak dataset of SRF in the genome scale, we did not identify a potential SRF binding element in *ItgaL*. Therefore, we focused on *ItgaM* and *Itgb2* in this study. This is explained in the revised manuscript.

The simple interpretation of this result is that the authors hypothesis, that Dia2 affects colonization through SRF control of integrin ITGB2 and ITGAM gene expression. The simplest interpretation is that in the DIA2 knockout, while ITGB2 expression drops, it does not become limiting, while expression of ITGAM does. They must be absolutely clear about this interpretation - toxicity is possible but not the simplest explanation. They still do not test ItgaL - this experiment should be done and the results shown.

We thank the reviewer for the comments. We agree with the reviewer about mDia2 on *Itgb2* and *ItgaM* and we made it clear in our revised manuscript.

Regarding *ItgaL*, as we pointed out in our response letter, the reasons that we did not test for *ItgaL* are that:

I: *ItgaL* was less down regulated compared to *ItgaM* and *Itgb2* when screened by real-time PCR in Figure 3G.

II. More important, when searching for the ChIP-seq binding peak dataset of SRF, we did not identify a potential SRF binding element in *ItgaL*.

There is also evidence that up-regulation of CD11a (encoded by *ItgaL*) in hematopoietic stem cells denotes the loss of long-term reconstitution potential, and all long-term reconstitution activity is within the CD11a negative fraction of bone marrow (Fathman et al, Stem Cell Reports 3, 707 (2014)). Therefore, it is very likely that over-expression of *ItgaL* in HSPCs could be harmful and do not rescue mDia2 null phenotypes.

Even if we could see some rescue effects of *ItgaL* on mDia2 deficiency, the mechanism of how mDia2 regulates the expression of *ItgaL* is still unclear since there is no clear SRF binding sites. We hope that the reviewer will agree with us.

- 2 *A critical point of the authors' model is that mDia2KO cells should be defective in recruitment of MAL to SRF binding sites (ANY SRF binding sites that recruitment MAL). This can be demonstrated by ChIP, but no data are provided. This is essential to validate the proposed integrin SRF binding sites, and for canonical target genes such as Acta2 or SRF itself.*

We agree with the reviewer and performed the experiments. We performed chromatin IP with antibodies against SRF, MAL followed by qRT-PCR of *ItgaM*, *Itgb2*, and *Acta2*. In new Figure 4H and S6B, we show enriched binding of SRF and MAL on *ItgaM*, *Itgb2* as well as *Acta2* gene locus upon serum stimulation.

This experiment goes some way towards supporting the idea that these genes are direct MAL targets, but it is surprising that the signal is only detected upon serum stimulation, since a significant proportion of resting cells have nuclear MAL, and would be expected to show binding (see Figure 3E). Comment please.

It is conventional practice to display ChIP-PCR data as "% input" - please amend

We thank the reviewer's comments. Binding in the resting cells would be an ideal result as the reviewer pointed out. However, several points need to be considered in this case:

(A) less than 20% cells have nuclear MAL at the baseline, which increased to more than 80% when serum was supplied (Figure 3E).

(B) With the above point being in mind, evidence from previous studies demonstrate that many SRF target genes can only be detected with significant binding peaks upon serum stimulation. As we explained in the previous letter, a previous study showed that 15% FCS dramatically stimulated the SRF binding compared with 0.3% FCS in a ChIP-sequencing assay (Genes Dev. 2014.28(9):943-58.) (see below). This is also the case in our revised Supplementary Figures 5B and 5C.

(C) With the low level of SRF activity at the resting stage, one has to consider the sensitivity of ChIP antibodies and variation of experiments. In this case, we do not see significant changes at the resting stage between cells from mDia2^{fl/fl} mice and those from mDia2^{fl/fl}Vav-Cre mice, especially using "% input" method to display the ChIP-PCR data. Therefore, we used fold enrichment method. The fold enrichment method is also widely accepted in the literature in that internal control is used to normalize the ChIP values to reduce experimental variations.

[Redacted]

- 3 *In this referees opinion better to reorder and keep Dia1 - but if Dia1 data is deleted must cite the Dia1 redundancy as unpublished observations in the Discussion to alert the reader.*

Thanks for the point. We noted in the discussion that mDia1 redundancy is unpublished observation.

- 4 *Figure 3E/F. The MAL and SRF and gene expression data is not convincing. 3E -The localization figure needs quantitation, it is only shown as images of representative cells, but it should also be quantified over 50-100 cells in Figure 3E, with markers for the cytoplasm.*

We repeated the immunofluorescent staining with a scaffolding component of the nuclear envelope lamin B1 (new supplementary figure S5A) and quantified the location of MAL in new Figure 3E. The cytoplasmic markers turned out to be difficult to use in HSPCs given the small amount of cytoplasm in these cells.

PLEASE give number of fields analyzed and number of cells / field, not %.

We thank the reviewer's suggestion. We revised the figure legend with the following information:

mDia2^{fl/fl} no FBS: N=71 cells from 17 random fields; mDia2^{fl/fl}Vav-Cre no FBS: N=80 cells from 15 random fields; mDia2^{fl/fl} with 10% FBS: N=56 cells from 12 random fields; mDia2^{fl/fl}Vav-Cre with 10% FBS: N=108 cells from 19 random fields.

5. *Figure 4 is not acceptable, and the authors appear to be unaware of the classical SRF literature.*

- Figure 4A. The SRE sequence consensus is CC(A/T)6GG, which should be referenced properly. There is no good fit to this consensus in either of these two sequences. What do the arrows indicate?

We included references of SRE in the revised manuscript. We agree with the reviewer that SRE on *Itgb2* intronic region does not perfectly match the classical SRE consensus sequence. However, the SRE on *ItgaM* does contain CC (A/T rich) GG motif [cc(ttgaga)gg], which was viewed as variants of the motif (Miano et al, Am J Physiol, Cell Physiol, 2007, 292(1) C70-81). Furthermore, recent studies using SRF ChIP-seq revealed SREs that do not contain CC(A/T)6GG consensus sequence, such as SP-1, ETS, and GFY as shown below (Genes & Dev. 2014. 28: 943-958). We discussed these points in the revised manuscript.

The experiment cited below looks at sequences associated with CHIPseq peaks, not direct DNA binding - there is no published evidence showing direct interaction between SRF and elements other than CC(A/T rich)GG.

[Redacted]

(Genes & Dev. 2014. 28: 943-958, Figure 2F)

The SRE-like sequence in *Itgb2* gene locus uncovered in our study was shown below, which is a reverse complementary sequence of the consensus CArG (CC(W6)GG, W=A or T), our binding sequence contain two deviations. One is located in the central region of W6 (labeled as *1), and the other one is the terminal G (labeled as *2).

(I). Studies classified CArG elements into two categories: consensus CArG and CArG like (Miano et al, Am J Physiol, Cell Physiol, 2007, 292(1) C70-81). The consensus CArG boxes follow the general rule, CCWWWWWWGG (W=A or T nucleotide). The CArG like sequences vary significantly, including:

Rule (A), central W substituted with either C or G (e.g., CCSW5GG, CCWSW4GG, CCWWSW3GG, etc), which supports the variation *1 in our CArG like sequence.

Rule (B), single nucleotide changes in the terminal sequences (either C being substituted with A, G, or T) or GG (either G may be substituted with A, C, or T), which supports the variation site of *2 in our CArG like sequence.

(II). Numerous studies supported that the CArG like DNA sequence physically bound to SRF. For example, the study from S.S.M. Rensen et al. / Cardiovascular Research 70 (2006) 136–145:

Figure 3A. Sequence comparison of CArG box-containing regions in the smoothelin-A and smoothelin-B promoters of human, mouse and rat. Dashed lines indicate probes used for Electrophoretic mobility shift assay (EMSA) experiments.

In this case, the CArG-far site had central G replacement, while the CArG-near sequence had a G substitution, both within W6 region. The physical binding of these CArG like elements with SRF has been confirmed by in vitro Electrophoretic mobility shift assay (EMSA) as shown in the following Figure 4.

[Redacted]

Fig. 4. A) Electrophoretic mobility shift assay of smoothelin-A CArG-far and CArG-near boxes and of the intronic SM-calponin CArG box (IC1 SM-calponin). Both smoothelin-A CArG boxes bound several nucleoprotein complexes, including a SRF-containing complex. Supershifted complexes are indicated by a left pointing arrow, SRF bound to CArG boxes is indicated by a right pointing arrow, additional specific complexes binding the smoothelin-A CArG boxes are indicated by diamonds.

Another nice example was from D.R. Wycuff et al. / *Virology* 324 (2004) 540–553, in which the viral CArG like element shared the same substitution at the same position as what we reported in our manuscript. The viral SRE binding with SRF has also been confirmed by EMSA in vitro.

[Redacted]

[Redacted]

6. *Figure 4F Assuming that the CCTTGAGAGG is a consensus, there is NO published literature supporting the notion that the C at position +6 downstream would have any effect on SRF binding – why was this mutant chosen? Where are two sites, and what is the evidence that they increase DNA binding activity?*

We found this site incidentally. This T site was spontaneously mutated to C in a PCR reaction in our template plasmid. Surprisingly, overexpression of this mutant hardly induce the luciferase expression. When this T/C mutation was corrected to T, the luciferase expression was restored. We included discussion in the revised manuscript.

(on lines 448...) The authors have no biochemical evidence that this change affects SRF-DNA interaction, and should say so. It is more likely that this change would affect another TF binding, perhaps one required for SRF to have its effect. Please amend.

We now include discussion that this change could affect another factor binding that could be required for SRF to have its effect.

7. Following the reviewer's suggestion, we performed both anti-SRF and anti-MAL ChIP assay using anti- GFP as a negative control. As we mentioned in our response to Major issue #2 from the reviewer, we captured specific enrichment of SRF and MAL binding on ItgaM, Itgb2 as well as Acta2 gene locus upon serum stimulation, which is absent with GFP antibody IP. Loss of mDia2 abolished these binding activities.

This experiment goes some way towards supporting the idea that these genes are direct MAL targets, but it is surprising that the signal is only detected upon serum stimulation, since a significant proportion of resting cells have nuclear MAL, and would be expected to show binding (see Figure 3E). Comment please.

Please see above our response to Point #2.

8 *Figure 4I. The CRISPR/Cas experiment is not adequately explained in the text or figure legends. It appears that there are multiple edits in the population (Figure 4D), yet a single population is shown for CD11B analysis - explain. Moreover, the effect of the deletion should be evaluated by qRT-PCR on ITGAM2, not by CD11 MFI.*

We thank the reviewer's comment, but we are not sure that we understand the reviewer's question. Figure 4D is the luciferase assay showing dose-dependent response by SRF overexpression and has nothing to do with editing. Following the reviewer's suggestion, we sorted the GFP/tRFP657 double positive LSK cells from sgLuc2p, sglTgaM intron2-T7, and sglTgaM intron2-T26 transduced cells in the transplant recipients and quantified the mRNA levels of ItgaM expression. As shown in new Figure 4I (right panel), genomic editing the SRE in ItgaM intronic region significantly reduced the transcripts of ItgaM. Figure 4I –

OK. But it is still completely unclear to me what the genotype of these knockouts are - there are multiple different edits shown in Figure S6 and the legends are inadequate. PLEASE fix.

We apologize for the unclear description. Hematopoiesis in the recipient mice was reconstituted by the transplanted Cas9 transgenic c-kit+ HSPCs transduced with sgRNA. The HSPCs with or without genomic editing will differentiate into lineage positive cells and be released into circulation. Thus, we obtained the peripheral blood from the recipient mice and performed PCR to amplify the target regions followed by sequencing to confirm the in-del produced by CRISPR-Cas9 and guide RNA (sgRNA). This routine has been extensively used in hematology research (Nat Biotechnol 32, 941-946 (2014)).

The editing by CRISPR-Cas9 usually won't be unique and uniformed. Our transplantation model is not the same as the established transgenic colony or mice. Therefore, it is not surprising that there are multiple different edits. However, regardless of different editing, they all disrupted the intact CArG-like motif, which reduces SRF-dependent transcriptional regulation of *ItgaM*.

To provide clear information and details, we revised the figure legend of Figure S6 as below:

(E-G) Determination of in vivo genome editing within *ItgaM* intron 2 region containing SRF binding element (CArG-like motif). The genomic DNA was isolated from the peripheral blood of the recipient mice and the target regions shown in the upper panel of E were PCR-amplified. The PCR products were cloned and subjected to sequencing to confirm the indels and estimate the editing frequencies. sgLuc2P, negative control, 100% (3/3) clones contain intact wide type sequence (lower panel in E); sglITGAM Intron 2_T7, 20% (1/5) clones contain a 7-bp nucleotide deletion occurred in the terminal region of CArG-like motif; 40% (2/5) clones contain a 361-bp deletion. 40% (2/5) clones contain a 193-bp deletion combined with a 19-bp insertion. They all lack the CArG-like motif (F); sglITGAM Intron 2_T26, 42.9% (3/7) clones contain a 644-bp deletion. 57.1% (4/7) clones contain a 439-bp deletion. Both lack the CArG-like motif (G).

Reviewer #3:

The authors have adequately addressed the comments from the reviewers and this very interesting study seems now acceptable for publication.

We thank the reviewer for these comments.

REVIEWERS' COMMENTS:

Reviewer #2 Remarks to the authors:

I think their response is OK. The one thing I would do is put the S7G figure in the main figure run as part of Figure 6, as an extension to 6H. As presented, the figure includes the experiment that worked, but not the one that didn't.

Reviewer #2 Remarks to the authors:

I think their response is OK. The one thing I would do is put the S7G figure in the main figure run as part of Figure 6, as an extension to 6H. As presented, the figure includes the experiment that worked, but not the one that didn't.

We thank the reviewer's comment. We have now put Figure S7G as new Figure 6i in the revised manuscript.